# ARC: A Generalist Graph Anomaly Detector with In-Context Learning

**Yixin Liu**[1,*]**, Shiyuan Li**[2,*]**, Yu Zheng**[3,*]**, Qingfeng Chen**[2,†]**, Chengqi Zhang**[4]**, Shirui Pan**[1,†]

[1]Griffith University, [2]Guangxi University, [3]La Trobe University,
[4]The Hong Kong Polytechnic University

yixin.liu@griffith.edu.au, shiy.li@alu.gxu.edu.cn, yu.zheng@latrobe.edu.au
qingfeng@gxu.edu.cn, chengqi.zhang@polyu.edu.hk, s.pan@griffth.edu.au

## Abstract

Graph anomaly detection (GAD), which aims to identify abnormal nodes that differ from the majority within a graph, has garnered significant attention. However, current GAD methods necessitate training specific to each dataset, resulting in high training costs, substantial data requirements, and limited generalizability when being applied to new datasets and domains. To address these limitations, this paper proposes ARC, a generalist GAD approach that enables a "one-for-all" GAD model to detect anomalies across various graph datasets on-the-fly. Equipped with in-context learning, ARC can directly extract dataset-specific patterns from the target dataset using few-shot normal samples at the inference stage, without the need for retraining or fine-tuning on the target dataset. ARC comprises three components that are well-crafted for capturing universal graph anomaly patterns: 1) smoothness-based feature **A**lignment module that unifies the features of different datasets into a common and anomaly-sensitive space; 2) ego-neighbor **R**esidual graph encoder that learns abnormality-related node embeddings; and 3) cross-attentive in-**C**ontext anomaly scoring module that predicts node abnormality by leveraging few-shot normal samples. Extensive experiments on multiple benchmark datasets from various domains demonstrate the superior anomaly detection performance, efficiency, and generalizability of ARC. The source code of ARC is available at https://github.com/yixinliu233/ARC.

## 1 Introduction

Graph anomaly detection (GAD) aims to distinguish abnormal nodes that show significant dissimilarity from the majority of nodes in a graph. GAD has broad applications across various real-world scenarios, such as fraud detection in financial transaction networks [1] and rumor detection in social networks [2]. As a result, GAD has attracted increasing research attention in recent years [3, 4, 5, 6, 7, 8]. Conventional GAD methods employ shallow mechanisms to model node-level abnormality [9, 10, 11]; however, they face limitations in handling high-dimensional features and complex interdependent relations on graphs. Recently, graph neural network (GNN)-based approaches have emerged as the go-to solution for the GAD problem due to their superior performance [4, 6]. Some GNN-based GAD approaches regard GAD as a supervised binary classification problem and use specifically designed GNN architectures to capture anomaly patterns [6, 12, 13, 14]. Another line of approaches targets the more challenging unsupervised paradigm, employing various unsupervised learning objectives and frameworks to identify anomalies without relying on labels [4, 15, 16, 17].

---

*Equal Contribution.

†Corresponding Authors.

38th Conference on Neural Information Processing Systems (NeurIPS 2024).

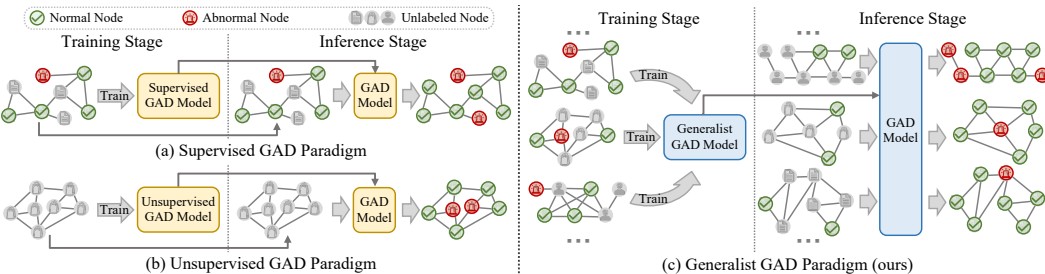

Figure 1: Sketch maps of (a) supervised, (b) unsupervised, and (c) generalist GAD paradigms.

Despite their remarkable detection performance, the existing GAD approaches follow a "**one model for one dataset**" learning paradigm (as shown in Fig. 1 (a) and (b)), necessitating dataset-specific training and ample training data to construct a detection model for each dataset. This learning paradigm inherently comes with the following limitations: ❶ *Expensive training cost.* For each dataset, we need to train a specialized GAD model from scratch, which incurs significant costs for model training, especially when dealing with large-scale graphs. ❷ *Data requirements.* Training a reliable GAD model typically needs sufficient in-domain data, sometimes requiring labels as well. The data requirements pose a challenge when applying GAD to scenarios with sparse data, data privacy concerns, or high label annotation costs. ❸ *Poor generalizability.* On a new-coming dataset, existing GAD methods require hyperparameter tuning or even model architecture modifications to achieve optimal performance, which increases the cost of applying them to new data and domains.

Given the above limitations, a natural question arises: *Can we train a "one-for-all" GAD model that can generalize to detect anomalies across various graph datasets from different application domains, without any training on the target data?* Following the trend of artificial general intelligence and foundation models, a new paradigm termed "**generalist anomaly detection**", originating from image anomaly detection, is a potential answer to this question [18]. As shown in Fig. 1 (c), in the generalist paradigm, we only need to train the GAD model once; afterward, the well-trained generalist GAD model can directly identify anomalies on diverse datasets, without any re-training or fine-tuning. Considering the diversity of graph data across different domains and datasets, the labels of *few-shot normal nodes* are required during the inference stage to enable the model to grasp the fundamental characteristics of the target dataset. Compared to conventional paradigms, the generalist paradigm eliminates the need for dataset-specific training, resulting in fewer computations, lower data costs, and stronger generalizability when applying GAD models to new datasets.

Nevertheless, due to the unique characteristics of graph data and GAD problem, it is non-trivial to design a generalist GAD approach. The challenge is three-fold: *C1 - Feature alignment.* Unlike image data, which are typically represented in a consistent RGB feature space, the feature dimensionality and semantic space can vary significantly across different graph datasets. Substituting features with unified representations generated by large language models may be a potential solution [19]; however, this approach is limited to specific feature semantics and cannot address more general cases [20]. *C2 - Representation encoding.* As the core of a generalist GAD model, a GNN-based encoder is expected to learn dataset-agnostic and abnormality-aware node embeddings for anomaly detection. However, in the absence of universal pre-trained foundation models [18] for graph data, crafting a potent encoder for a generalist GAD model presents a challenge. *C3 - Few-shot sample-guided prediction.* Existing GAD methods typically focus on single dataset settings, where dataset-specific knowledge is embedded in the model through training, enabling it to predict abnormality for each node independently. In contrast, a generalist GAD model should derive such knowledge from a small number of normal nodes. In this case, how to effectively utilize the few-shot normal samples during inference remains an open question.

To tackle these challenges, we introduce ARC, a generalist GAD approach based on in-context learning. ARC comprises three meticulously designed modules, each targeting a specific challenge. To address *C1*, we introduce a smoothness-based feature **A**lignment module, which not only standardizes features across diverse datasets to a common dimensionality but also arranges them in an anomaly-sensitive order. To deal with *C2*, we design an ego-neighbor **R**esidual graph encoder. Equipped with a multi-hop residual-based aggregation scheme, the graph encoder learns attributes that indicate high-order affinity and heterophily, capturing informative and abnormality-aware embeddings across different datasets. Last but not least, to solve *C3*, we propose a cross-attentive in-**C**ontext anomaly scoring module. Following the in-context learning schema, we treat the few-shot normal nodes as context samples and utilize a cross-attention block to reconstruct the embeddings of unlabeled

samples based on the context samples. Then, the reconstruction distance can serve as the anomaly score for each unlabeled node. In summary, this paper makes the following contributions:

- **Problem.** We, for the first time, propose to investigate the generalist GAD problem, aiming to detect anomalies from various datasets with a single GAD model, without dataset-specific fine-tuning.
- **Methodology.** We propose a novel generalist GAD method ARC, which can detect anomalies in new graph datasets on-the-fly via in-context learning based on few-shot normal samples.
- **Experiments.** We conduct extensive experiments to validate the anomaly detection capability, generalizability, and efficiency of ARC across multiple benchmark datasets from various domains.

## 2 Related Work

In this section, we offer a brief review of pertinent related works, with a more extensive literature review available in Appendix A.

**Anomaly Detection.** Anomaly detection (AD) aims to identify anomalous samples that deviate from the majority of samples [21]. Mainstream AD methods focus on unsupervised settings and employ various unsupervised techniques to build the models [22, 23, 24, 25, 26, 27, 28, 29]. To enhance the generalizability of AD methods across diverse datasets, RegAD [30] considers few-shot setting and trains a single generalizable model capable of being applied to new in-domain data without re-training or fine-tuning. WinCLIP [31] utilizes visual-language models (VLMs, e.g., CLIP [32]) with well-crafted text prompts to perform zero/few-shot AD for image data. InCTRL [18], as the first generalist AD approach, integrates in-context learning and VLMs to achieve domain-agnostic image AD with a single model. However, due to their heavy reliance on pre-trained vision encoders/VLMs and image-specific designs, these approaches excel in AD for image data but face challenges when applied to graph data.

**Graph Anomaly Detection (GAD).** In this paper, we focus on the node-level AD on graphs and refer to it as "graph anomaly detection (GAD)" following [6, 33, 34]. While shallow methods [9, 10, 11] show limitations in handling complex real-world graphs [4], the advanced approaches are mainly based on GNNs [35]. The GNN-based approaches can be divided into supervised and unsupervised approaches [3, 5, 7]. Supervised GAD approaches assume that the labels of both normal and anomalous nodes are available for model training [7]. Hence, related studies mainly introduce GAD methods in a binary classification paradigm [6, 12, 13, 14, 36, 37]. In contrast, unsupervised GAD approaches do not require any labels for model training. They employ several unsupervised learning techniques to learn anomaly patterns on graph data, including data reconstruction [4, 34, 38], contrastive learning [15, 39, 40], and other auxiliary objectives [16, 17, 41, 42]. Nevertheless, all the above methods adhere to the conventional paradigm of "one model for one dataset". Although some GAD approaches [43, 44] can handle cross-domain scenarios, their requirement for high correlation (e.g., aligned node features) between source and target datasets limits their generalizability. Differing from existing methods, our proposed ARC is a "one-for-all" GAD model capable of identifying anomalies across target datasets from diverse domains, without the need for re-training or fine-tuning.

**In-Context Learning (ICL).** ICL enables a well-trained model to be effectively (fine-tuning-free) adapted to new domains, datasets, and tasks based on minimal in-context examples (a.k.a. context samples), providing powerful generalization capability of large language models (LLMs) [45, 46, 47] and computer vision (CV) models [18, 48, 49, 50]. Two recent approaches, PRODIGY [51] and UniLP [52] attempt to use ICL for GNNs to solve the node classification and link prediction tasks, respectively. However, how to use ICL to deal with the generalist GAD problem where only normal context samples are available still remains open.

## 3 Problem Statement

**Notations.** Let $\mathcal{G} = (\mathcal{V}, \mathcal{E}, \mathbf{X})$ be an attributed graph with $n$ nodes and $m$ edges, where $\mathcal{V} = \{v_1, \cdots, v_n\}$ and $\mathcal{E}$ are the set of nodes and edges, respectively. The node-level attributes are included by feature matrix $\mathbf{X} \in \mathbb{R}^{n \times d}$, where each row $\mathbf{X}_i$ indicates the feature vector for node $v_i$. The inter-node connectivity is represented by an adjacency matrix $\mathbf{A} \in \{0, 1\}^{n \times n}$, where the $i, j$-th entry $\mathbf{A}_{ij} = 1$ means $v_i$ and $v_j$ are connected and vice versa.

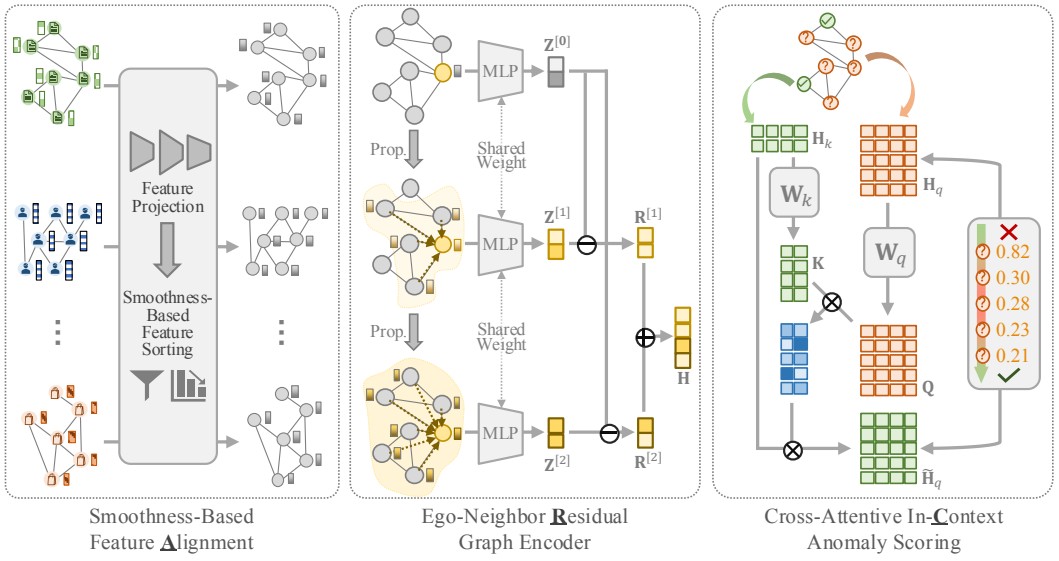

Figure 2: The overall pipeline of ARC, the proposed generalist GAD approach.

**Conventional GAD Problem.** GAD aims to differentiate abnormal nodes $\mathcal{V}_a$ from normal nodes $\mathcal{V}_n$ within a given graph $\mathcal{G} = (\mathcal{V}, \mathcal{E}, \mathbf{X})$, where $\mathcal{V}_a$ and $\mathcal{V}_n$ satisfy $\mathcal{V}_a \cup \mathcal{V}_n = \mathcal{V}$, $\mathcal{V}_a \cap \mathcal{V}_n = \emptyset$, and $|\mathcal{V}_a| \ll |\mathcal{V}_n|$. An anomaly label vector $\mathbf{y} \in \{0, 1\}^n$ can be used to denote the abnormality of each node, where the $i$-th entry $\mathbf{y}_i = 1$ *iff* $v \in \mathcal{V}_a$ and $\mathbf{y}_i = 0$ *iff* $v \in \mathcal{V}_n$. Formally, the goal of GAD is to learn an anomaly scoring function (i.e., GAD model) $f : \mathcal{V} \to \mathbb{R}$ such that $f(v') > f(v)$ for $\forall v' \in \mathcal{V}_a$ and $\forall v \in \mathcal{V}_n$. In the conventional GAD setting of "one model for one dataset", the GAD model $f$ is optimized on the target graph dataset $\mathcal{D} = (\mathcal{G}, \mathbf{y})$ with a subset of anomaly labels (in supervised setting) or without labels (in unsupervised setting). After sufficient training, the model $f$ can identify anomalies within the target graph $\mathcal{G}$ during the inference phase.

**Generalist GAD Problem.** In this paper, we investigate the ***generalist GAD problem***, wherein we aim to *develop a generalist GAD model capable of detecting abnormal nodes across diverse graph datasets from various application domains without any training on the specific target data*. Formally, we define the generalist GAD setting, aligning it with its counterpart in image AD as introduced by Zhu et al. [18]. Specifically, let $\mathcal{T}_{train} = \{\mathcal{D}_{train}^{(1)}, \cdots, \mathcal{D}_{train}^{(N)}\}$ be a collection of training datasets, where each $\mathcal{D}_{train}^{(i)} = (\mathcal{G}_{train}^{(i)}, \mathbf{y}_{train}^{(i)})$ is a labeled dataset from an arbitrary domain. We aim to train a generalist GAD model $f$ on $\mathcal{T}_{train}$, and $f$ is able to identify anomalies within any test graph dataset $\mathcal{D}_{test}^{(i)} \in \mathcal{T}_{test}$, where $\mathcal{T}_{test} = \{\mathcal{D}_{test}^{(1)}, \cdots, \mathcal{D}_{test}^{(N')}\}$ is a collection of testing datasets. Note that $\mathcal{T}_{train} \cap \mathcal{T}_{test} = \emptyset$ and the datasets in $\mathcal{T}_{train}$ and $\mathcal{T}_{test}$ can be drawn from different distributions and domains. Following [18], we adopt a "normal few-shot" setting during inference: for each $\mathcal{D}_{test}^{(i)}$, only a handful of $n_k$ normal nodes ($n_k \ll n$) are available, and the model $f$ is expected to predict the abnormality of the rest nodes without re-training and fine-tuning.

## 4  ARC: A generalist GAD approach

In this section, we introduce ARC, a generalist GAD approach capable of identifying anomalies across diverse graph datasets without the need for specific fine-tuning. The overall pipeline of ARC is demonstrated in Fig. 2. Firstly, to align the features of different datasets, we introduce a *smoothness-based feature alignment* module (Sec. 4.1), which not only projects features onto a common plane but also sorts the dimensions in an anomaly-sensitive order. Next, to capture abnormality-aware node embeddings, we propose a simple yet effective GNN model termed *ego-neighbor residual graph encoder* (Sec. 4.2), which constructs node embeddings by combining residual information between an ego node and its neighbors. Finally, to leverage knowledge from few-shot context samples for predicting node-level abnormality, we introduce a *cross-attentive in-context anomaly scoring* module (Sec. 4.3). Using the cross-attention block, the model learns to reconstruct query node embeddings based on context node embeddings. Ultimately, the drift distance between the original and reconstructed query embeddings can quantify the abnormality of each node.

## 4.1 Smoothness-Based Feature Alignment

Graph data from diverse domains typically have different features, characterized by differences in dimensionality and unique meanings for each dimension. For example, features in a citation network usually consist of textual and meta-information associated with each paper, whereas in a social network, the features may be the profile of each user. Therefore, in the first step, we need to align the features into a shared feature space. To achieve this, we introduce the feature alignment module in ARC, consisting of two phases: feature projection, which aligns dimensionality, and smoothness-based feature sorting, which reorders features according to their smoothness characteristics.

**Feature Projection.** At the first step of ARC, we employ a feature projection block to unify the feature dimensionality of multiple graph datasets [20]. Specifically, given a feature matrix $\mathbf{X}^{(i)} \in \mathbb{R}^{n^{(i)} \times d^{(i)}}$ of $\mathcal{D}^{(i)} \in \mathcal{T}_{train} \cup \mathcal{T}_{test}$, the feature projection is defined by a linear mapping:

$$\tilde{\mathbf{X}}^{(i)} \in \mathbb{R}^{n^{(i)} \times d_u} = \mathrm{Proj}\left(\mathbf{X}^{(i)}\right) = \mathbf{X}^{(i)}\mathbf{W}^{(i)}, \tag{1}$$

where $\tilde{\mathbf{X}}^{(i)}$ is the projected feature matrix for $\mathcal{D}^{(i)}$, $d_u$ is a predefined projected dimension shared across all datasets, and $\mathbf{W}^{(i)} \in \mathbb{R}^{d^{(i)} \times d_u}$ is a dataset-specific linear projection weight matrix. To maintain generality, $\mathbf{W}^{(i)}$ can be defined using commonly used dimensionality reduction approaches such as singular value decomposition [53] (SVD) and principal component analysis [54] (PCA).

**Smoothness-Based Feature Sorting.** Although feature projection can align dimensionality, the semantic meaning of each projected feature across different datasets remains distinct. Considering the difficulty of semantic-level matching without prior knowledge and specific fine-tuning [19, 20], in this paper, we explore an alternative pathway: aligning features based on their contribution to anomaly detection tasks. Through analytical and empirical studies, we pinpoint that *the smoothness of each feature is strongly correlated with its contribution to GAD*. Building on this insight, in ARC, we propose to sort the features according to their contribution as our alignment strategy.

From the perspective of graph signal processing, Tang et al. [6] have demonstrated that the inverse of the low-frequency energy ratio monotonically increases with the anomaly degree. In other words, high-frequency graph signals tend to play a more significant role in detecting anomalies. Similar findings have also been observed from the standpoint of spatial GNNs [37, 55], where heterophily information has been shown to be crucial in discriminating anomalies. Motivated by these findings, can we develop a met-

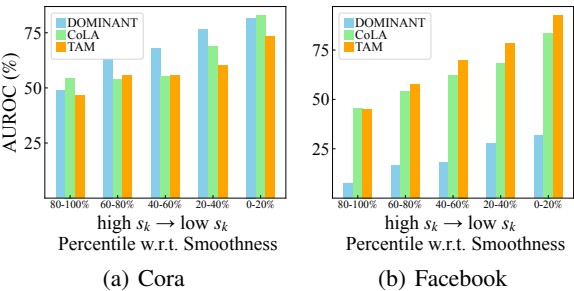

Figure 3: AUROC on data with 5 groups of features.

ric to gauge the contribution of each feature to GAD based on its frequency/heterophily? Considering its correlation to frequency [56] and heterophily [57, 58, 59, 60], in this paper, we select feature-level **smoothness** as the measure for contribution. Formally, given a graph $\mathcal{G} = (\mathcal{V}, \mathcal{E}, \mathbf{X})$ with a normalized feature matrix $\mathbf{X}$, the smoothness of the $k$-th feature dimension is defined as:

$$s_k(\mathbf{X}) = -\frac{1}{|\mathcal{E}|} \sum_{(v_i, v_j) \in \mathcal{E}} \left(\mathbf{X}_{ik} - \mathbf{X}_{jk}\right)^2, \tag{2}$$

where a lower $s_k$ indicates a significant change in the $k$-th feature between connected nodes, implying that this feature corresponds to a high-frequency graph signal and exhibits strong heterophily.

To verify whether smoothness can indicate the contribution of features in GAD, we further conduct empirical analysis (experimental setup and more results can be found in Appendix B). Concretely, we sort the raw features of each dataset based on the smoothness $s_k$ and divide them into 5 groups according to the percentile of $s_k$. Then, we train different GAD models using each group of features separately, and the performance is shown in Fig. 3 and 8. On both datasets, a model-agnostic observation is that the features with lower $s_k$ are more helpful in discriminating anomalies. The consistent trend demonstrates the effectiveness of $s_k$ as an indicator of the role of features in GAD.

In light of this, given the *projected features* of different datasets, we can align their feature spaces by rearranging the permutation of features based on the descending order of $s_k$ w.r.t. each projected feature. For all datasets, the feature in the first column is the one with the lowest $s_k$, which deserves more attention by ARC; conversely, features with less contribution (i.e. higher $s_k$) are placed at the end. In this way, the GNN-based model can learn to filter graph signals with different smoothness levels automatically and predict anomalies accordingly. During inference, the smoothness-related information remains transferable because we adhere to the same alignment strategy.

## 4.2 Ego-Neighbor Residual Graph Encoder

Once the features are aligned, we employ a GNN-based graph encoder to learn node embeddings that capture both semantic and structural information for each node. The learned embedding can be utilized to predict the abnormality of the corresponding node with the downstream anomaly scoring module. A naive solution is directly employing commonly used GNNs, such as GCN [61] or GAT [62], as the graph encoder. However, due to their low-pass filtering characteristic, these GNNs face difficulty in capturing abnormality-related patterns that are high-frequency and heterophilic [6, 37]. Moreover, most GNNs, including those tailored for GAD, tend to prioritize capturing node-level semantic information while disregarding the affinity patterns of local subgraphs [17]. Consequently, employing existing GNN models as the encoder may overemphasize dataset-specific semantic knowledge, but overlook the shared anomaly patterns (i.e. local node affinity) across different datasets.

To address the above issues, we design an ego-neighbor residual graph encoder for ARC. Equipped with a residual operation, the encoder can capture multi-hop affinity patterns of each node, providing valuable and comprehensive information for anomaly identification. Similar to the "propagation then transformation" GNN architecture in SGC [63], our graph encoder consists of three steps: multi-hop propagation, shared MLP-based transformation, and ego-neighbor residual operation. In the first two steps, we perform propagation on the aligned feature matrix $\mathbf{X}' = \mathbf{X}^{[0]}$ for $L$ iterations, and then conduct transformation on the initial and propagated features with a shared MLP network:

$$\mathbf{X}^{[l]} = \tilde{\mathbf{A}}\mathbf{X}^{[l-1]}, \quad \mathbf{Z}^{[l]} = \mathrm{MLP}\left(\mathbf{X}^{[l]}\right), \tag{3}$$

where $l \in \{0, \cdots, L\}$, $\mathbf{X}^{[l]}$ is the propagated feature matrix at the $l$-th iteration, $\mathbf{Z}^{[l]}$ is the transformed representation matrix at the $l$-th iteration, and $\tilde{\mathbf{A}}$ is the normalized adjacency matrix [61, 63]. Note that, unlike most GNNs that only consider the features/representations after $L$-iter propagation, here we incorporate both the initial features and intermediate propagated features and transform them into the same representation space. After obtaining $\mathbf{Z}^{[0]}, \cdots, \mathbf{Z}^{[L]}$, we calculate the residual representations by taking the difference between $\mathbf{Z}^{[l]}$ ($1 \leq l \leq L$) and $\mathbf{Z}^{[0]}$, and then concatenate the multi-hop residual representations to form the final embeddings:

$$\mathbf{R}^{[l]} = \mathbf{Z}^{[l]} - \mathbf{Z}^{[0]}, \quad \mathbf{H} = [\mathbf{R}^{[1]}||\cdots||\mathbf{R}^{[L]}], \tag{4}$$

where $\mathbf{R}^{[l]}$ is the residual matrix at the $l$-th iteration, $\mathbf{H} \in \mathbb{R}^{n \times d_e}$ is the output embedding matrix, and $||$ denotes the concatenation operator.

**Discussion.** Compared to existing GNNs, our graph encoder offers the following advantages. Firstly, with the residual operation, the proposed encoder emphasizes the difference between the ego node and its neighbors rather than ego semantic information. This approach allows for the explicit modeling of local affinity through the learned embeddings. Since local affinity is a crucial indicator of abnormality and this characteristic can be shared across diverse datasets [17], the learned embeddings can offer valuable discriminative insights for downstream prediction. Second, the residual operation performs as a high-pass filter on the graph data, aiding ARC in capturing more abnormality-related attributes, i.e., high-frequency signals and local heterophily. Moreover, unlike existing approaches [15, 17] that only consider 1-hop affinity, our encoder also incorporates higher-order affinity through the multi-hop residual design, which enables ARC to capture more complex graph anomaly patterns. More discussion and comparison to the existing GNNs/GAD methods are conducted in Appendix C.

## 4.3 Cross-Attentive In-Context Anomaly Scoring

To utilize the few-shot normal samples (denoted by **context nodes**) to predict the abnormality of the remaining nodes (denoted by **query nodes**), in ARC, we devise an in-context learning module with a cross-attention mechanism for anomaly scoring. The core idea of our in-context learning module

is to reconstruct the node embedding of each query node using a cross-attention block to blend the embeddings of context nodes. Then, the drift distance between the original and reconstructed embeddings of a query node can serve as the indicator of its abnormality.

Specifically, we partition the embedding matrix $\mathbf{H}$ into two parts by indexing the corresponding row vectors: the embeddings of context nodes $\mathbf{H}_k \in \mathbb{R}^{n_k \times d_e}$ and the embeddings of query nodes $\mathbf{H}_q \in \mathbb{R}^{n_q \times d_e}$. Then, a cross-attention block is utilized to reconstruct each row of $\mathbf{H}_q$ through a linear combination of $\mathbf{H}_k$:

$$\mathbf{Q} = \mathbf{H}_q \mathbf{W}_q, \quad \mathbf{K} = \mathbf{H}_k \mathbf{W}_k, \quad \tilde{\mathbf{H}}_q = \text{Softmax}\left(\frac{\mathbf{Q}\mathbf{K}^\top}{\sqrt{d_e}}\right)\mathbf{H}_k, \quad (5)$$

where $\mathbf{Q} \in \mathbb{R}^{n_q \times d_e}$ and $\mathbf{K} \in \mathbb{R}^{n_k \times d_e}$ are the query and key matrices respectively, $\mathbf{W}_q$ and $\mathbf{W}_k$ are learnable parameters, and $\tilde{\mathbf{H}}_q$ is the reconstructed query embedding matrix. Note that, unlike the conventional cross-attention blocks [64, 65, 66] that further introduce a value matrix $\mathbf{V}$, our block directly multiplies the attention matrix with $\mathbf{H}_k$. This design ensures that $\tilde{\mathbf{H}}_q$ is in the same embedding space as $\mathbf{H}_q$ and $\mathbf{H}_k$. Thanks to this property, given a query node $v_i$, we can calculate its anomaly score $f(v_i)$ by computing the L2 distance between its query embedding vector $\mathbf{H}q_i$ and the corresponding reconstructed query embedding vector $\tilde{\mathbf{H}}q_i$, i.e., $f(v_i) = d(\mathbf{H}q_i, \tilde{\mathbf{H}}q_i) = \sqrt{\sum_{j=1}^{d_e}\left(\mathbf{H}q_{ij} - \tilde{\mathbf{H}}q_{ij}\right)^2}$.

**Discussion.** The design of cross-attentive in-context anomaly scoring follows a basic assumption: normal query nodes have similar patterns to several context nodes, and hence their embeddings can be easily represented by the linear combination of context node embeddings. Consequently, given a normal node, its original and reconstructed embeddings can be close to each other in the embedding space. In contrast, abnormal nodes may display distinct patterns compared to normal ones, making it difficult to depict their corresponding abnormal query embeddings using context embeddings. As a result, their drift distance $s_i$ can be significantly larger. Fig. 4 provides examples for the scenarios of (a) single-class normal and (b) multi-class normal[3]. In both cases, the drift distance ($\rightarrow$) can be

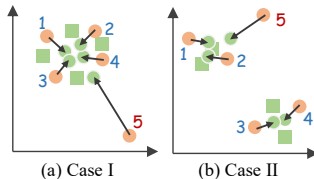

(a) Case I      (b) Case II

Figure 4: Toy examples of query embeddings (●), reconstructed query embeddings (●), and context embeddings (■).

a significant indicator for distinguishing anomaly (5) from normal nodes (1~4). Interestingly, if the attention matrix assigns uniform weights to all context nodes, then our scoring module becomes a one-class classification model [22]. This property ensures the anomaly detection capability of ARC even without extensive training. A detailed discussion is conducted in Appendix D.2.

**Model Training.** To optimize ARC on training datasets $\mathcal{T}_{train}$, we employ a marginal cosine similarity loss to minimize the drift distance of normal nodes while maximizing the drift distance of abnormal nodes. Specifically, given graph data with anomaly labels, we randomly select $n_k$ normal nodes as context nodes and sample an equal number of normal and abnormal nodes as query nodes. Then, given a query node $v_i$ with embedding $\mathbf{H}q_i$, reconstructed embedding $\tilde{\mathbf{H}}q_i$, and anomaly label $\mathbf{y}_i$, the sample-level loss function can be written by:

$$\mathcal{L} = \begin{cases} 1 - \cos\left(\mathbf{H}q_i, \tilde{\mathbf{H}}q_i\right), & \text{if } \mathbf{y}_i = 0 \\ \max\left(0, \cos\left(\mathbf{H}q_i, \tilde{\mathbf{H}}q_i\right) - \epsilon\right), & \text{if } \mathbf{y}_i = 1 \end{cases} \quad (6)$$

where $\cos(\cdot, \cdot)$ and $\max(\cdot, \cdot)$ denote the cosine similarity and maximum operation, respectively, and $\epsilon$ is a margin hyperparameter. Detailed algorithmic description and complexity analysis of ARC can be found in Appendix E.

## 5   Experiments

### 5.1   Experimental Setup

**Datasets.** To learn generalist GAD models, we train the baseline methods and ARC on a group of graph datasets and test on another group of datasets. For comprehensive evaluations, we consider

---

[3]Specific definitions see Appendix D.1

Table 1: Anomaly detection performance in terms of AUROC (in percent, mean±std). Highlighted are the results ranked first, second, and third. "Rank" indicates the average ranking over 8 datasets.

| Method | Cora | CiteSeer | ACM | BlogCatalog | Facebook | Weibo | Reddit | Amazon | Rank |
|---|---|---|---|---|---|---|---|---|---|
| *Supervised - Pre-Train Only* | | | | | | | | | |
| GCN | $59.64_{\pm8.30}$ | $60.27_{\pm8.11}$ | $60.49_{\pm9.65}$ | $56.19_{\pm6.39}$ | $29.51_{\pm4.86}$ | $76.64_{\pm17.69}$ | $50.43_{\pm4.41}$ | $46.63_{\pm3.47}$ | 8.9 |
| GAT | $50.06_{\pm2.65}$ | $51.59_{\pm3.49}$ | $48.79_{\pm2.73}$ | $50.40_{\pm2.80}$ | $51.88_{\pm2.16}$ | $53.06_{\pm7.48}$ | $51.78_{\pm4.04}$ | $50.52_{\pm17.22}$ | 10.0 |
| BGNN | $42.45_{\pm11.57}$ | $42.32_{\pm11.82}$ | $44.00_{\pm13.69}$ | $47.67_{\pm8.52}$ | $54.74_{\pm25.29}$ | $32.75_{\pm35.35}$ | $50.27_{\pm3.84}$ | $52.26_{\pm3.31}$ | 11.1 |
| BWGNN | $54.06_{\pm3.27}$ | $52.61_{\pm2.88}$ | $67.59_{\pm0.70}$ | $56.34_{\pm1.21}$ | $45.84_{\pm4.97}$ | $53.38_{\pm1.61}$ | $48.97_{\pm5.74}$ | $55.26_{\pm16.95}$ | 9.0 |
| GHRN | $59.89_{\pm6.57}$ | $56.04_{\pm9.19}$ | $55.65_{\pm6.37}$ | $57.64_{\pm3.48}$ | $44.81_{\pm8.06}$ | $51.87_{\pm14.18}$ | $46.22_{\pm2.33}$ | $49.48_{\pm17.13}$ | 9.8 |
| *Unsupervised - Pre-Train Only* | | | | | | | | | |
| DOMINANT | $66.53_{\pm1.15}$ | $69.47_{\pm2.02}$ | $70.08_{\pm2.34}$ | $74.25_{\pm0.65}$ | $51.01_{\pm0.78}$ | $92.88_{\pm0.32}$ | $50.05_{\pm4.92}$ | $48.94_{\pm2.69}$ | 5.8 |
| CoLA | $63.29_{\pm8.88}$ | $62.84_{\pm9.52}$ | $66.85_{\pm4.43}$ | $50.04_{\pm3.25}$ | $12.99_{\pm11.68}$ | $16.27_{\pm5.64}$ | $52.81_{\pm6.69}$ | $47.40_{\pm7.97}$ | 9.5 |
| HCM-A | $54.28_{\pm4.73}$ | $48.12_{\pm6.80}$ | $53.70_{\pm4.64}$ | $55.31_{\pm0.57}$ | $35.44_{\pm13.97}$ | $65.52_{\pm12.58}$ | $48.79_{\pm2.75}$ | $43.99_{\pm0.72}$ | 11.4 |
| TAM | $62.02_{\pm2.39}$ | $72.27_{\pm0.83}$ | $74.43_{\pm1.59}$ | $49.86_{\pm0.73}$ | $65.88_{\pm6.66}$ | $71.54_{\pm0.18}$ | $55.43_{\pm0.33}$ | $56.06_{\pm2.19}$ | 5.6 |
| *Unsupervised - Pre-Train & Fine-Tune* | | | | | | | | | |
| DOMINANT | $72.23_{\pm0.34}$ | $74.69_{\pm0.32}$ | $74.34_{\pm0.12}$ | $74.61_{\pm0.04}$ | $49.92_{\pm0.55}$ | $92.21_{\pm0.10}$ | $52.14_{\pm5.06}$ | $59.06_{\pm2.80}$ | 3.6 |
| CoLA | $67.62_{\pm4.26}$ | $70.75_{\pm3.42}$ | $69.11_{\pm0.67}$ | $62.49_{\pm3.38}$ | $64.70_{\pm18.86}$ | $31.55_{\pm6.02}$ | $58.12_{\pm0.67}$ | $52.51_{\pm6.66}$ | 5.4 |
| HCM-A | $56.45_{\pm4.93}$ | $55.54_{\pm4.07}$ | $57.69_{\pm3.59}$ | $55.10_{\pm0.29}$ | $36.57_{\pm10.72}$ | $71.89_{\pm2.79}$ | $49.15_{\pm2.72}$ | $42.20_{\pm0.55}$ | 10.1 |
| TAM | $62.56_{\pm2.10}$ | $76.54_{\pm1.33}$ | $86.29_{\pm1.57}$ | $57.69_{\pm0.88}$ | $76.26_{\pm3.70}$ | $71.73_{\pm0.16}$ | $56.62_{\pm0.49}$ | $57.13_{\pm1.59}$ | 3.4 |
| *Ours* | | | | | | | | | |
| ARC | $87.45_{\pm0.74}$ | $90.95_{\pm0.59}$ | $79.88_{\pm0.28}$ | $74.76_{\pm0.06}$ | $67.56_{\pm1.60}$ | $88.85_{\pm0.14}$ | $60.04_{\pm0.69}$ | $80.67_{\pm1.81}$ | 1.5 |

graph datasets spanning a variety of domains, including social networks, citation networks, and e-commerce co-review networks, each of them with either injected anomalies or real anomalies [7, 15, 17]. Inspired by [52], we train the models on the largest dataset of each type and conduct testing on the remaining datasets. Specifically, the training datasets $\mathcal{T}_{train}$ comprise PubMed, Flickr, Questions, and YelpChi, while the testing datasets $\mathcal{T}_{test}$ consist of Cora, CiteSeer, ACM, BlogCatalog, Facebook, Weibo, Reddit, and Amazon. For detailed information, please refer to Appendix F.1.

**Baselines.** We compare ARC with both supervised and unsupervised methods. Supervised methods include two conventional GNNs, i.e., GCN [61] and GAT [62], and three state-of-the-art GNNs specifically designed for GAD, i.e., BGNN [67], BWGNN [6], and GHRN [37]. Unsupervised methods include four representative approaches with distinct designs, including the generative method DOMINANT [4], the contrastive method CoLA [15], the hop predictive method HCM-A [16], and the affinity-based method TAM [17]. For detailed information, refer to Appendix F.2.

**Evaluation and Implementation.** Following [7, 17, 68], we employ AUROC and AUPRC as our evaluation metrics for GAD. We report the average AUROC/AUPRC with standard deviations across 5 trials. We train ARC on all the datasets in $\mathcal{T}_{train}$ jointly, and evaluate the model on each dataset in $\mathcal{T}_{test}$ in an in-context learning manner ($n_k = 10$ as default). For the supervised baselines, we follow the same training and testing procedure (denoted as "pre-train only"), since no labeled anomaly is available for fine-tuning. For the unsupervised baselines, we consider two settings: "pre-train only" and "pre-train & fine-tune". In the latter, we additionally conduct dataset-specific fine-tuning with a few epochs. To standardize the feature space in baseline methods, we utilize either learnable projection or random projection as an adapter between the raw feature and the model input layer. We utilize random search to determine the optimal hyperparameters for both the baselines and ARC. Since our goal is to train generalist GAD models, we do not perform dataset-specific hyperparameter search, but instead use the same set of hyperparameters for all testing datasets. More implementation details can be found in Appendix F.3.

## 5.2 Experimental Results

**Performance Comparison.** Table 1 shows the comparison results of ARC with baseline methods in terms of AUROC. Results in AUPRC are provided in Appendix G.1. We have the following observations. ❶ ARC demonstrates strong anomaly detection capability in the generalist GAD scenario, without any fine-tuning. Specifically, ARC achieves state-of-the-art performance on 5 out of 8 datasets and demonstrates competitive performance on the remainder. On several datasets, ARC demonstrates significant improvement compared to the best baseline (e.g., ↑21.1% on Cora, ↑18.8% on CiteSeer, and ↑36.6% on Amazon). ❷ Simply

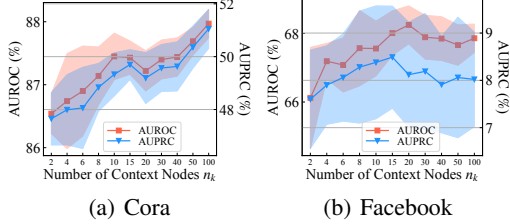

(a) Cora          (b) Facebook

Figure 5: Performance with varying $n_k$.

pre-training the dataset-specific GAD methods typically results in poor generalization capability to new datasets. Specifically, the AUROC of the majority of "pre-train only" approaches is close to random guessing (50%) or even lower. ❸ With dataset-specific fine-tuning, the baseline methods achieve better performance in the majority of cases. However, the improvement can be minor or even negative in some cases, demonstrating the limitations of fine-tuning. Additionally, we observe that their performance is sometimes lower than the reported results from training models from scratch [4, 15, 17], indicating potential risk of negative transfer within the "pre-train & fine-tune" paradigm. ❹ Unsupervised baselines (except HCM-A) generally outperform the supervised ones, highlighting the difficulty of training a generalist GAD model using the binary classification paradigm.

**Effectiveness of #Context Nodes.** To investigate how the number of context nodes $n_k$ affects the performance of ARC during inference, we vary $n_k$ within the range of 2 to 100. The results are shown in Fig. 5 (more results are in Appendix G.2). From the figure, we observe that the performance of ARC increases as more context nodes are involved, demonstrating its capability to leverage these labeled normal nodes with in-context learning. Furthermore, we can conclude that ARC is also label-efficient: when $n_k \geq 10$, the performance gain from using more context nodes becomes minor; moreover, even when $n_k$ is extremely small, ARC can still perform well on the majority of datasets.

**Ablation Study.** To verify the effectiveness of each component of ARC, we make corresponding modifications to ARC and designed three variants: 1) **w/o A**: using random projection to replace smoothness-based feature alignment; 2) **w/o R**: using GCN to replace ego-neighbor residual graph encoder; and 3) **w/o C**: using binary classification-based predictor and loss to replace cross-

Table 2: AUROC of ARC and its variants.

| Variant | Cora | Cite. | ACM | Blog. | Face. | Wei. | Red. | Ama. |
|---|---|---|---|---|---|---|---|---|
| ARC | 87.45 | 90.95 | 79.88 | 74.76 | 67.56 | 88.85 | 60.04 | 80.67 |
| w/o A | 80.65 | 83.35 | 79.29 | 73.86 | 62.80 | 89.69 | 54.60 | 64.76 |
| w/o R | 37.44 | 31.52 | 61.83 | 49.30 | 20.38 | 97.72 | 52.94 | 50.15 |
| w/o C | 47.39 | 53.98 | 54.24 | 60.46 | 48.86 | 42.84 | 51.03 | 69.02 |

attentive in-context anomaly scoring. The results are demonstrated in Table 2 (full results are in Appendix G.3). From the results, we can conclude that all three components significantly contribute to the performance. Among them, the in-context anomaly scoring module has a significant impact, as the performance of **w/o C** is close to random guessing on most datasets. The residual graph encoder also has a significant impact on the final performance. Notably, Weibo dataset is an exception where the GCN encoder performs better. A possible reason is that the Weibo dataset exhibits different anomaly patterns compared to the others.

**Efficiency Analysis.** To assess the runtime efficiency of ARC, we compare the inference and fine-tuning time on the ACM dataset. As depicted in Fig. 6, ARC demonstrates comparable runtime performance with the fastest GNNs (e.g., GCN and BWGNN), and significantly outperforms the unsupervised methods in terms of efficiency. Additionally, we observe that dataset-specific fine-tuning consumes more time compared to inference.

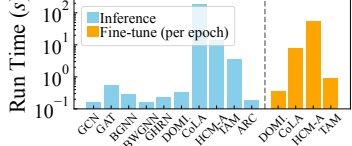

Figure 6: Time comparison.

**Visualization.** To investigate the weight allocation mechanism of the cross-attention module in ARC, we visualize the attention weights between context nodes and query nodes in Fig.7 (additional results are in Appendix G.4). From Fig. 7(a), it is evident that ARC tends to assign uniform attention weights to normal nodes, leading to reconstructed embeddings that closely resemble the average embedding of the context nodes. Conversely, anomalies are reconstructed using a combination of 1 or 2 context nodes, suggesting that their embeddings are farther from the center. This allocation aligns with the case of "single-class normal" in Fig. 4(a). Differently, in Fig. 7(b), we observe that each normal query node is assigned to several context nodes following two fixed patterns, corresponding to the case of "multi-class normal" in Fig. 4(b). In summary, the cross-attention module enables ARC to adapt to various normal/anomaly distribution patterns, enhancing its generalizability.

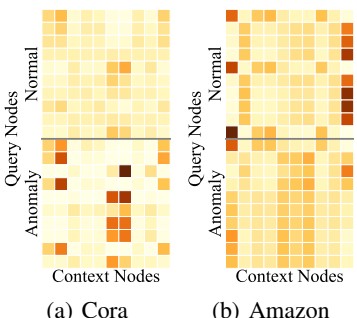

(a) Cora    (b) Amazon

Figure 7: Visualization results.

# 6 Conclusion

In this paper, we take the first step towards addressing the generalist GAD problem, aiming to detect anomalies across diverse graph datasets with a "one-for-all" GAD model, without requiring dataset-specific fine-tuning. We introduce ARC, a novel and well-crafted in-context learning-based generalist GAD approach, capable of identifying anomalies on-the-fly using only few-shot normal nodes. Extensive experiments on real-world datasets from various domains demonstrate the detection prowess, generalizability, and efficiency of ARC compared to existing approaches. One limitation is that ARC can only use normal context samples during inference but cannot directly utilize abnormal context samples, even when they are available. A potential future direction could involve developing generalist GAD methods that utilize context samples containing both anomalies and normal instances.

## Acknowledgments and Disclosure of Funding

This research was partly funded by Australian Research Council (ARC) under grants FT210100097 and DP240101547 and the CSIRO – National Science Foundation (US) AI Research Collaboration Program.

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

# A Detailing Related Work

**Anomaly Detection.** The objective of anomaly detection (AD) is to identify anomalous samples that deviate from the majority of samples [21]. Due to the difficulty of collecting labeled anomaly data, mainstream AD methods mainly focus on unsupervised settings. To capture anomaly patterns without guidance by annotated labels, existing studies employ several advanced techniques to learn powerful AD models, such as one-class classification [22, 23], distance measurement [24, 69], data reconstruction [25, 70, 71], generative models [26, 72, 73], and self-supervised learning [27, 74]. For example, DeepSVDD [22] introduces a fully deep one-class classification objective for unsupervised anomaly detection, optimizing a data-enclosing hypersphere in output space to extract common factors of variation and demonstrating theoretical properties such as the $v$-property. AnoDDPM [72], a simplex noise-based approach for anomaly detection, enhances anomaly capture with a partial diffusion strategy [75] and multiscale simplex noise processing, facilitating faster inference and training on high-resolution images. While effective, these approaches are tailored to identify abnormal samples within a predetermined target dataset (i.e., the dataset for training), restricting their generalizability to new domains.

**Cross-Dataset Anomaly Detection.** Recently, some advanced AD methods aim to transcend dataset limitations, enhancing their generalizability across diverse datasets. A research line aims to address the AD problem under domain or distribution shifts [76, 77, 78, 79]; however, these approaches require domain relevance between the source and target datasets, thus limiting their generalizability. To enable the model to better understand the patterns in target datasets, a viable approach is to incorporate a few-shot setting, allowing access to a limited number of normal samples from the target datasets. Under the few-shot setting, RegAD [30] is a pioneering approach that trains a single generalizable model capable of being applied to new in-domain data without re-training or fine-tuning. WinCLIP [31] utilizes visual-language models (VLMs, e.g., CLIP [32]) with well-crafted text prompts to perform zero/few-shot AD for image data. InCTRL [18], as the first generalist AD approach, integrates in-context learning and VLMs to achieve domain-agnostic image AD with a single model. However, due to their heavy reliance on pre-trained vision encoders/VLMs and image-specific designs, these approaches excel in anomaly detection for image data but face challenges when applied to graph data.

**Anomaly Detection on Graph Data.** Based on the granularity of anomaly samples within graph data, existing AD approaches can be primarily categorized into three classes: node-level [4, 6], edge-level [80, 81, 82], and graph-level [83, 84, 85, 86, 87, 88] AD. Due to its broad real-world applications, node-level AD receives the most research attention [3]. In this paper, we focus on the node-level AD and refer to it as "graph anomaly detection (GAD)", following the convention of most previous papers [6, 33, 34].

Early GAD methods aimed to detect anomalies through shallow mechanisms. For example, AMEN [9] detects anomalies by utilizing the attribute correlations of nodes in each ego-network on the attribute network. In addition, residual analysis is another common method to measure the anomalies of nodes on an attribute network. In particular, Radar [10] characterizes the residuals of attribute information and their coherence with network information for anomaly detection. Further, ANOMALOUS [11] proposes joint learning of attribute selection and anomaly detection based on CUR decomposition and residual analysis. Despite the success of these methods on low-dimensional attribute graph data, they do not work well when graphs have complex structures and high-dimensional attributes [89, 90, 91, 92] due to the limitations of their shallow mechanisms.

To overcome the limitations of shallow approaches, recently, GNN-based methods have become the de facto solution for GAD tasks. Existing GNN-based GAD approaches can be divided into two research lines: supervised GAD and unsupervised GAD [3, 5, 7]. Supervised GAD approaches assume that the labels of both normal and anomalous nodes are available for model training [7]. Hence, related studies mainly focus on improving graph convolutional operators, model architectures, and supervised objective functions, to leverage labels to learn node-level anomaly patterns [6, 12, 13, 14, 36, 37]. For example, the spatial GNN has been redesigned mainly in terms of message passing and aggregation mechanisms. In particular, considering that GNN-based fraud detectors fail to effectively identify fraudsters in disguise, CARE-GNN [12] uses a label-aware similarity metric and introduces reinforcement learning to aggregate selected neighbors with different relationships to counteract disguises. Concurrently, spectral GNNs, relate graphical anomalies to high-frequency spectral distributions. Tang et al. [6] analyzed anomalies for the first time from the perspective

of the spectral spectrum of the graph and proposed the Beta wavelet GNN (BWGNN), which has spectrally and spatially localized band-pass filters to better capture anomalies. In addition, anomalies are usually associated with high-frequency components in the spectral representation of a graph. Therefore, GHRN [37] prunes inter-class edges by emphasizing the high-frequency components of the graph, which can effectively isolate the anomalous nodes and thus obtain better anomaly detection performance.

In contrast, unsupervised GAD approaches do not require any labels for model training. Similar to unsupervised AD for image data, unsupervised GAD approaches employ several unsupervised learning techniques to learn anomaly patterns on graph data, including data reconstruction [4, 34, 38], contrastive learning [15, 39, 40], and other auxiliary objectives [16, 17, 41, 42, 93]. For example, DOMINANT [4] is a reconstruction-based approach that employs a graph convolution autoencoder to reconstruct both the adjacency and attribute matrices simultaneously, assessing node abnormality through a weighted sum of the reconstruction error terms. Similarly, ComGA [34] is a community-aware attribute GAD framework based on tailored GCNs to capture local, global, and structural anomalies. CoLA [15], the first contrastive self-supervised learning framework for GAD, samples novel contrast instance pairs in an unsupervised manner, utilizing contrastive learning to capture local information. HCM-A [16] integrates both local and global contextual information, employs hop count prediction as a self-supervised task, and utilizes Bayesian learning to enhance anomaly identification. TAM [17] optimizes the proposed anomaly metric (affinity) on the truncated graph end-to-end, considering one-class homophily and local affinity. Nevertheless, all the above methods adhere to the conventional paradigm of "one model for one dataset".

Although some GAD approaches [43, 44] can handle cross-domain scenarios, their requirement for high correlation (e.g., aligned node features) between source and target datasets limits their generalizability. Differing from existing methods, our proposed ARC is a "one-for-all" GAD model capable of identifying anomalies across target datasets from diverse domains, without the need for re-training or fine-tuning.

**In-Context Learning.** In-context learning (ICL) can be effectively adapted to new tasks based on minimal in-context examples, providing a powerful generalization capability of large language models (LLMs) [45, 46, 47]. For example, Brown et al. [45] demonstrate the remarkable ability of language models to perform diverse tasks with minimal training examples. Through few-shot learning, these models exhibit robustness and adaptability across various domains, exhibiting their potential as general-purpose learners in natural language processing (NLP). Further, given that pre-training objectives are not specifically optimized for ICL [94], Min et al. introduced MetaICL [95] as a solution to bridge the divide between pre-training and downstream ICL utilization. This method involves continuously training the pre-trained LLMs across diverse tasks using demonstration examples, thereby enhancing its few-shot capabilities. In contrast, supervised in-context fine-tuning [94] suggests constructing self-supervised training data aligned with the ICL format to utilize the original corpora for warm-up in downstream tasks. They converted raw text into input-output pairs and investigated four self-supervised objectives, such as masked token prediction and classification tasks.

ICL has also generated research attention in the field of computer vision (CV), where it has been widely used in vision tasks by designing specialized discretization tokens as prompts [18, 48, 49, 50]. For instance, Chen et al. [48] proposed to use a unified interface to represent the output of each task as a sequence of discrete tokens, the neural network can be trained for a variety of tasks using a single model architecture and loss function, thus eliminating the need for customization for specific tasks. Furthermore, given a shot prompt as a task description, the sequence output adapts to the prompt to generate task-specific results. Differently, Amir et al. [96] proposes an innovative method for in-context visual prompting by framing a wide range of vision tasks as grid in-painting problems. This approach leverages image in-painting to generate visual prompts, guiding models to complete various vision tasks by filling in missing parts of an image based on contextual information, thereby enhancing adaptability and performance across different visual challenges.

Very recently, few studies apply ICL to graph learning and GNNs. As an initial attempt to use ICL for GNNs, PRODIGY [51] leverages a prompt graph-based framework to conduct few-shot ICL for node-level classification and edge-level prediction tasks. Further, UniLP [52] introduces ICL to resolve conflicting connectivity patterns caused by distributional differences between different graphs. Nevertheless, these methods require context samples in multiple classes for ICL during inference.

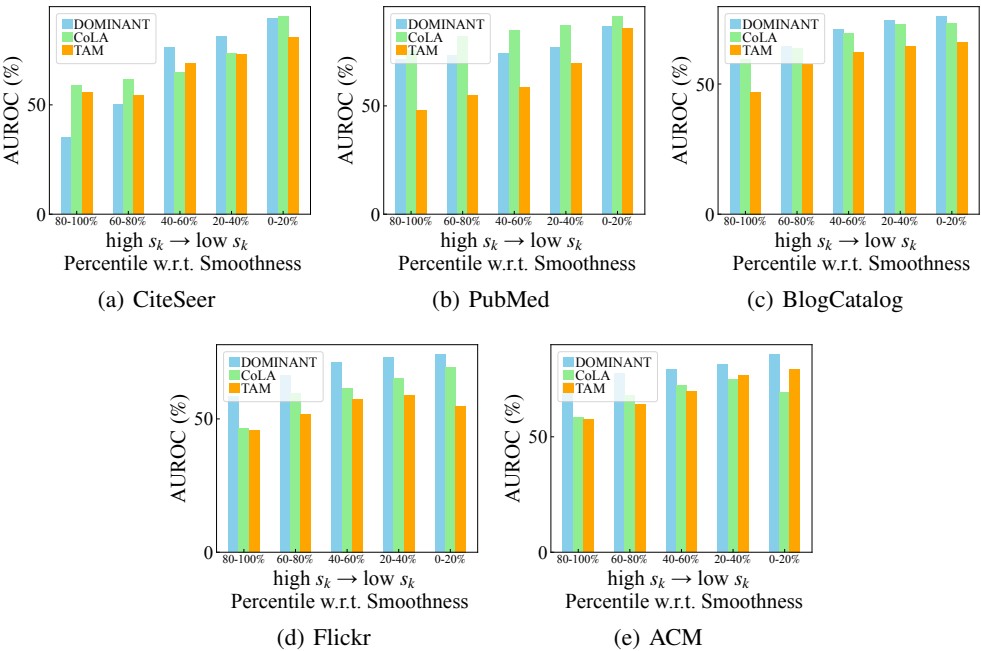

Figure 8: AUROC on data with 5 groups of features split by $s_k$.

Then, how to leverage one-class context samples with ICL in the scenario of generalist GAD remains an open question.

## B    Motivated Experiments for Smoothness-Based Feature Sorting

**Experimental Setup.** To verify whether **smoothness** can indicate the contribution of features to GAD, we conducted motivated experiments. We consider three mainstream GAD methods: DOMINANT [4], CoLA [15], and TAM [17]. For each method, we conduct the experiments with hyper-parameters reported in the original article across all datasets. At the data pre-processing phase, we first calculate $s_k$ based on the raw features and sort them in descending order. According to the new order, we divide the features into 5 groups of feature subsets, denoted by the percentile of 80%-100%, 60%-80%, $\cdots$, 0%-20%. Subsequently, the five feature subsets are sequentially used as inputs for different methods, and the methods are trained to obtain their AUROC values. We statistic the results of 5 random experiments for each method and reported the average AUROC.

**Results.** Additional experimental results are shown in Fig. 8. As we can witness in the figure, in most cases, features with lower $s_k$ are observed to be more helpful in distinguishing anomalies on most of the datasets. This consistent model-independent trend indicates the effectiveness of $s_k$ as an indicator of the contribution of each feature in GAD.

## C    Discussion of Ego-Neighbor Residual Graph Encoder

In this section, we discuss the connections and differences between the ego-neighbor residual graph encoder in ARC (R-ENC for short) and the existing GNNs and GAD methods.

**R-ENC v.s. Radar [10].** Radar, one of the representative shallow GAD methods, also leverages a residual-based mechanism for anomaly detection. Our proposed R-ENC differs from Radar in the following key aspects:

- Radar calculates the residual on the original feature space, which inevitably suffers from high complexity and less representative capability on high-dimensional graph data. In contrast, R-ENC computes residuals based on the output embeddings of a GNN, which enables more efficient processing and enhances the representative power of the model. This

approach allows R-ENC to better capture the underlying structure and anomalies within the graph data.

- In Radar, the summation of residual $\mathbf{R}$ directly serves as the anomaly score for each node, where the irrelevant information within several residual entries may hinder the model's accuracy and effectiveness. Differently, in ARC, we specifically use R-ENC to generate the embedding of each node, and employ another learnable scoring module to calculate the anomaly score based on these embeddings. In this case, our method can effectively filter out noise and irrelevant features, allowing the scoring module to focus on the most informative aspects of the node embeddings.

- Radar employs the Laplacian matrix to calculate the residual, meaning that only first-order residual information is considered. In contrast, our R-ENC incorporates multi-hop residual aggregation, enhancing its ability to detect subtle anomalies by considering both local and global graph structures through high-order residuals.

**R-ENC v.s. ResGCN [97].** ResGCN is a GNN-based GAD approach with a residual-based mechanism. Similar to Radar, ResGCN also uses the summation of residual $\mathbf{R}$ as the anomaly score. However, the inclusion of irrelevant information within several residual entries can impair the model's accuracy and effectiveness. Moreover, ResGCN employs a two-branch design, with the node representation by GCN and residual information by MLP calculated by two different network modules. Compared to R-ENC with simpler designs, ResGCN has lower operational efficiency.

**R-ENC v.s. CoLA [15].** CoLA learns the anomaly patterns by maximizing the agreement between the embedding of each node and its neighboring nodes. Specifically, CoLA employs a bilinear module to compute the agreement score. Such an agreement can also be modeled by the first-order residual in R-ENC, which is computed by the difference between the ego embedding and the 1-hop aggregated embeddings. Compared to CoLA, a significant advantage of R-ENC is its ability to capture not only first-order residuals but also high-order residuals. This makes R-ENC a more robust and comprehensive encoder for graph anomaly detection, addressing a wider range of anomaly patterns across different datasets.

**R-ENC v.s. TAM [17].** TAM utilizes local affinity, i.e., the feature-level or embedding-level similarity between a node and its neighbors, as the indicator of each node's abnormality. R-ENC, with its residual operation, can also capture local affinity. Specifically, by computing the first-order residual between an ego node and its 1-hop neighbors, the local affinity can be indicated by the negative summation of the first-order residuals since they are highly correlated. Again, R-ENC can not only capture the first-order affinity but also the high-order affinity with the multi-hop residual operator.

**R-ENC v.s. Heterophily-aware GNNs [37].** Existing studies indicate that a key solution to handle the GAD problem is to enable heterophily-aware graph convolutional operation with high-pass filtering [37]. A feasible filter is graph Laplacian, whose normalized and self-loop added version can be written by $\mathbf{L} = \mathbf{I} - \tilde{\mathbf{A}}$. For R-ENC, its first-order residual can also be viewed as a Laplacian-based graph convolution. Specifically, if we simplify the MLP-based feature transformation as a weight matrix $\mathbf{W}$, the ego information can be written by $\mathbf{Z}^{[0]} = \mathbf{XW}$, while the representation $\mathbf{Z}^{[1]} = \tilde{\mathbf{A}}\mathbf{XW}$. Then, the first-order residual can be written by:

$$\mathbf{R}^{[1]} = \mathbf{Z}^{[1]} - \mathbf{Z}^{[0]} = \tilde{\mathbf{A}}\mathbf{XW} - \mathbf{XW} = -\mathbf{LXW}. \tag{7}$$

That is to say, the first-order residual in R-ENC can be regarded as Laplacian-based high-pass filtering (note that the negative sign can be fused into the learnable weight $\mathbf{W}$). Such a nice property enables ARC to capture high-frequency graph signals and heterophily information through residual-based embeddings, thereby enhancing its capability to detect anomalies in diverse and complex graph structures.

## D  Discussion of Cross-Attentive In-Context Anomaly Scoring

### D.1  Definitions of Single-Class and Multi-Class Normal

**Dataset with single-class normal.** In this type of dataset, the normal samples share the same pattern or characteristics. For example, in a network traffic monitoring system dataset, normal behavior

might be defined by regular patterns of data packets exchanged between a specific set of IP addresses. Any deviation from this single, well-defined pattern, such as an unexpected spike in data volume or communication with unknown IP addresses, can be flagged as anomalous.

**Dataset with multi-class normal.** In this type of dataset, the normal samples are divided into multiple classes, each with distinct patterns or characteristics. For example, in a corporate email communication network dataset, normal data might be defined by regular patterns of email exchanges within specific departments, such as HR, IT, and Finance. Any deviation from these well-defined patterns, such as a sudden spike in emails between normally unconnected departments or an unusual volume of emails from an individual employee to external addresses, can be detected as anomalous.

## D.2 ARC as One-Class Classification model

In this subsection, we first introduce the basic definition of one-class classification (OC) model, and then discuss the connection between one-class classification and cross-attentive in-context anomaly scoring module (C-AS for short).

**One-class classification.** The core idea of OC model is to measure the abnormality of each sample according to the distance between its representation $\mathbf{h}$ and a center representation $\mathbf{c}$ [22]. Here $\mathbf{c}$ can be a fixed random representation vector or dynamically adjusted as the mean of all samples' representation vectors. Formally, the anomaly score by OC model can be written by:

$$f(\mathbf{x}_i) = \|\phi(\mathbf{x}_i) - \mathbf{c}\|^2 = \|\mathbf{h}_i - \mathbf{c}\|^2,\tag{8}$$

where $\phi(\cdot)$ is a neural network model, as defined in Deep SVDD [22]. Intuitively, a normal sample tends to have a similar representation to the majority of samples, and hence the distance between its representation and $\mathbf{c}$ should be closer.

**C-AS as an OC model.** In C-AS, we use a cross-attention block to calculate the weighted sum of context embeddings $\mathbf{H}_k$ into $\tilde{\mathbf{H}}_{q_i}$ for a query node $q_i$. For an initialized model, we assume that the parameters $\mathbf{W}_q$ and $\mathbf{W}_k$ are random enough, making $\mathbf{Q}$ and $\mathbf{K}$ become uniform noise matrices. In this case, each entry in the attention matrix $\mathbf{T} = \mathrm{Softmax}\left(\frac{\mathbf{Q}\mathbf{K}^\top}{\sqrt{d_e}}\right)$ can be $\frac{1}{n_k}$, indicating that the attention matrix assigns uniform weights to all context nodes for all query nodes. Then, all the reconstructed query embeddings are equal to the average embedding of context nodes:

$$\tilde{\mathbf{H}}_{q_1} = \cdots = \tilde{\mathbf{H}}_{q_{n_q}} = \frac{1}{n_k}\mathbf{1}^T\mathbf{H}_k.\tag{9}$$

Since the average context embedding is the center embedding of a group of few-shot normal samples, we can naturally define the center embedding $\mathbf{c} = \frac{1}{n_k}\mathbf{1}^T\mathbf{H}_k$. Recalling that we define the anomaly score as the L2 distance between $\tilde{\mathbf{H}}_{q_i}$ and $\mathbf{H}_{q_i}$, then for all query nodes, the anomaly scoring can be rewritten by:

$$f(v_i) = d(\mathbf{H}q_i, \tilde{\mathbf{H}}q_i) = d(\mathbf{H}q_i, \mathbf{c}) = \|\mathbf{H}_{q_i} - \mathbf{c}\|^2.\tag{10}$$

That is to say, the C-AS module serves as an OC model under random initialization. Note that in practice, the attention matrix cannot be so ideal, but it can still assign relevantly average weights for the context embeddings. Such merit ensures that ARC can perform like an OC model, effectively detecting anomalies in the case of single-class normal (Fig. 4 (a)) even without costly training.

**C-AS goes beyond OC model.** Thanks to its inherent mechanism, the OC model can effectively handle single-class normal scenarios. However, in the case of multi-class normals (Fig. 4 (b)), a single center is not sufficient to model multiple normal class centers. Unlike the OC model, C-AS can address this issue through cross-attention. Specifically, the cross-attention block learns to reconstruct a query embedding by assigning higher weights to several (but not all) context embeddings that are close to the query node. This way, for a query node, the cross-attention block can automatically learn the center of its corresponding normal class, rather than simply using the average context embedding. The awareness of multiple normal classes ensures that ARC can handle both single-class and multi-class normal cases.

---

**Algorithm 1:** Smoothness-based Feature Alignment

---

**Input:** Graph $\mathcal{G}$.
**Parameters:** Projected dimension $d_u$.

**1** Extract $\mathbf{X}$, $\mathcal{E}$, and $\mathcal{V}$ from $\mathcal{G}$
**2** $\tilde{\mathbf{X}} \in \mathbb{R}^{n \times d_u} \leftarrow$ Calculate projected features by linear projection via Eq. (1)
**3** **for** $k = 1 : d_u$ **do**
**4** $\quad \mid \quad s_k \leftarrow$ Calculate feature-level smoothness of the $k$-th column of $\tilde{\mathbf{X}}$ via Eq. (2)
**5** **end**
**6** $\mathbf{X}' \leftarrow$ Rearrange the permutation of features of $\tilde{\mathbf{X}}$ based on the descending order of $\mathbf{s}$
**7** Return $\mathcal{G} = (\mathcal{V}, \mathcal{E}, \mathbf{X}')$

---

---

**Algorithm 2:** The Training algorithm of ARC

---

**Input:** Training datasets $\mathcal{T}_{train}$.
**Parameters:** Number of epoch $E$; Propagation iteration: $L$.

**1** Initialize model parameters
**2** **for** $\mathcal{D}^{(i)} \in \mathcal{T}_{train}$ **do**
**3** $\quad \mid \quad$ Align features in $\mathcal{G}^{(i)}$ via Algo. 1
**4** **end**
**5** **for** $e = 1 : E$ **do**
**6** $\quad \mid \quad$ **for** $\mathcal{D}^{(i)} \in \mathcal{T}_{train}$ **do**
**7** $\quad \mid \quad \mid \quad$ Obtain $\mathbf{X}^{(i)'}, \mathcal{E}^{(i)}, \mathcal{V}^{(i)}, \mathbf{y}^{(i)}$ from $\mathcal{D}^{(i)}$
**8** $\quad \mid \quad \mid \quad$ **for** $l = 1 : L$ **do**
**9** $\quad \mid \quad \mid \quad \mid \quad$ $\mathbf{Z}^{(i),[l]} \leftarrow$ Propagate and transform $\mathbf{X}^{(i)'} = \mathbf{X}^{(i),[0]}$ via Eq. (3)
**10** $\quad \mid \quad \mid \quad \mid \quad$ $\mathbf{R}^{(i),[l]} \leftarrow$ Calculate residual of $\mathbf{Z}^{(i),[l]}$ via Eq. (4)
**11** $\quad \mid \quad \mid \quad$ **end**
**12** $\quad \mid \quad \mid \quad$ $\mathbf{H}^{(i)} \leftarrow$ Concatenate $[\mathbf{R}^{(i),[1]} || \cdots || \mathbf{R}^{(i),[L]}]$ via Eq. (4)
**13** $\quad \mid \quad \mid \quad$ $\mathbf{H}_q^{(i)}, \mathbf{H}_k^{(i)} \leftarrow$ Randomly split query and context node sets and indexing from $\mathbf{H}^{(i)}$
**14** $\quad \mid \quad \mid \quad$ $\tilde{\mathbf{H}}^{(i)} \leftarrow$ Calculate cross attention from $\mathbf{H}_q^{(i)}, \mathbf{H}_k^{(i)}$ via Eq. (5)
**15** $\quad \mid \quad \mid \quad$ Calculate loss $\mathcal{L}$ from $\tilde{\mathbf{H}}_q^{(i)}, y_q^{(i)}$ via Eq. (6)
**16** $\quad \mid \quad \mid \quad$ Update model parameters via gradient descent.
**17** $\quad \mid \quad$ **end**
**18** **end**

---

# E   Algorithm and Complexity

## E.1   Algorithmic description

The algorithmic description of the feature alignment in ARC, the training process of ARC, and inference process of ARC are summarized in Algo. 1, Algo. 2, and Algo. 3, respectively.

## E.2   Complexity Analysis

In the testing phase, the time complexity consists of two main components: feature alignment and model inference. For feature alignment, the overall complexity is $\mathcal{O}(ndd_u + d_u m + d_u log(d_u))$, where $m = |E|$ is the number of edges. Here, the first term is used for feature projection, while the second and third terms are used for smoothness computation and feature reordering, respectively. The model inference is divided into two main parts: embedding generation and anomaly scoring. The complexity of node embedding generation is $\mathcal{O}(L(md_u + nd_u h + nh^2))$, where the first term is used for feature propagation and the rest of the terms are used for residual encoding by MLP. The anomaly scoring, on the other hand, mainly involves cross-attention computation with time complexity of $\mathcal{O}(n_q n_k h + n_q h)$, where $n_q$ is the number of query nodes and $n_k$ is the number of context nodes.

**Algorithm 3:** The Inference algorithm of ARC

**Input:** Test dataset $\mathcal{D}$ with few-shot normal nodes $\{v_{k_1}, \cdots, v_{k_{n_k}}\}$.
**Parameters:** Well-trained model weight parameters.
1 Align features in $\mathcal{G}$ via Algo. 1
2 Obtain $\mathbf{X}', \mathcal{E}, \mathcal{V}$ from $\mathcal{G}$
3 **for** $l = 1 : L$ **do**
4     $\mathbf{Z}^{[l]} \leftarrow$ Propagate and transform $\mathbf{X}^{(i)'} = \mathbf{X}^{[0]}$ via Eq. (3)
5     $\mathbf{R}^{[l]} \leftarrow$ Calculate residual of $\mathbf{Z}^{[l]}$ via Eq. (4)
6 **end**
7 $\mathbf{H} \leftarrow$ Concatenate $[\mathbf{R}^{[1]}||\cdots||\mathbf{R}^{[L]}]$ via Eq. (4)
8 $\mathbf{H}_q, \mathbf{H}_k \leftarrow$ Separate query and context node sets and indexing from $\mathbf{H}$
9 $\tilde{\mathbf{H}}_q \leftarrow$ Calculate cross attention from $\mathbf{H}_q, \mathbf{H}_k$ via Eq. (5)
10 $\mathbf{d} \leftarrow$ Computing the L2 distance between $\tilde{\mathbf{H}}_q$ and $\mathbf{H}_q$
11 Return $\mathbf{d}$ as the anomaly scores $f(\cdot)$ for query nodes

Table 3: The statistics of datasets.

| Dataset | Train | Test | #Nodes | #Edges | #Features | Avg. Degree | #Anomaly | %Anomaly |
|---|---|---|---|---|---|---|---|---|
| Citation network with injected anomalies | | | | | | | | |
| Cora | - | ✓ | 2,708 | 5,429 | 1,433 | 3.90 | 150 | 5.53 |
| CiteSeer | - | ✓ | 3,327 | 4,732 | 3,703 | 2.77 | 150 | 4.50 |
| ACM | - | ✓ | 16,484 | 71,980 | 8,337 | 8.73 | 597 | 3.62 |
| PubMed | ✓ | - | 19,717 | 44,338 | 500 | 4.50 | 600 | 3.04 |
| Social network with injected anomalies | | | | | | | | |
| BlogCatalog | - | ✓ | 5,196 | 171,743 | 8,189 | 66.11 | 300 | 5.77 |
| Flickr | ✓ | - | 7,575 | 239,738 | 12,047 | 63.30 | 450 | 5.94 |
| Social network with real anomalies | | | | | | | | |
| Facebook | - | ✓ | 1,081 | 55,104 | 576 | 50.97 | 25 | 2.31 |
| Weibo | - | ✓ | 8,405 | 407,963 | 400 | 48.53 | 868 | 10.30 |
| Reddit | - | ✓ | 10,984 | 168,016 | 64 | 15.30 | 366 | 3.33 |
| Questions | ✓ | - | 48,921 | 153,540 | 301 | 3.13 | 1,460 | 2.98 |
| Co-review network with real anomalies | | | | | | | | |
| Amazon | - | ✓ | 10,244 | 175,608 | 25 | 17.18 | 693 | 6.76 |
| YelpChi | ✓ | - | 23,831 | 49,315 | 32 | 2.07 | 1,217 | 5.10 |

## F    Details of Experimental Setup

### F.1    Description of Datasets

In total, we considered 12 benchmark datasets. We divide the datasets into 4 groups: ❶ citation network with injected anomalies, ❷ social network with injected anomalies, ❸ social network with real anomalies, and ❹ co-review network with real anomalies. Within each type, we consider the largest dataset as one of the training datasets, and the rest datasets as the testing datasets. The detailed statistics of the datasets are shown in Table 3. These datasets are selected from different domains and with injected or real anomalies to ensure that our proposed ARC model learns extensive anomaly patterns. The diversity of the above data can maximally ensure that ARC can effectively adapt to new and unseen graphs. Specifically, the detailed descriptions for the datasets are given as follows:

- **Cora, CiteSeer, PubMed [98], and ACM [99]** are four citation networks. In these datasets, nodes represent scientific publications, while edges denote the citation links between them. Each publication is characterized by a bag-of-words representation for its node attribute vector, with the dimensionality determined by the size of the respective dictionary.

- **BlogCatalog and Flickr [4, 100]** stand as typical social blog directories, facilitating user connections through following relationships. Each user is depicted as a node, with inter-node links symbolizing mutual following. Node attributes encompass the personalized textual content generated by users within social network, such as blog posts or shared photos with tag descriptions.

- **Amazon and YelpChi [101, 102]** are datasets about the relationship between users and reviews. Amazon is designed to identify users paid to write fake reviews for products, and three different graph datasets are derived from Amazon using different types of relations to construct adjacency matrix [17, 103]. YelpChi aims to identify anomalous reviews on Yelp.com that unfairly promote or demote products or businesses. Based on [101, 104], three different graph datasets derived from Yelp using different connections in user, product review text, and time. In this work, we focus on Amazon-UPU (users who have reviewed at least one of the same product) and YelpChi-RUR (reviews posted by the same user).

- **Facebook [105]** is a social network in which users can build relationships with others and share their friends.

- **Reddit [106]** serves as a forum posts network sourced from the social media platform Reddit, where users labeled as banned are identified as anomalies. Textual content from posts is transformed into vectors to serve as node attributes.

- **Weibo [106]** dataset encompasses a graph of users and their associated hashtags from the Tencent Weibo platform. Within a defined temporal window (e.g., 60 seconds), consecutive posts by a user are labeled as potentially suspicious behavior. Users engaging in a minimum of five such instances are classified as "suspicious". The raw feature vector includes the location of a micro-blog post and bag-of-words features.

- **Questions [107]** dataset originates from Yandex Q, a platform dedicated to question-answering. Users represent the nodes, while the connections between them signify the presence or absence of a question-and-answer interaction within a one-year timeframe. The node features are derived from the average of the FastText embeddings of the words in the user description, with an additional binary feature indicating users without description.

**Anomaly Injection.** For the datasets with injected anomalies, we use the strategy introduced in [4, 15] to inject anomalous nodes. Specifically, we inject a set of anomaly combinations for each dataset by perturbing the topology and node attributes, respectively [108]. In terms of structural perturbation, this is done by generating small cliques of otherwise unrelated nodes as anomalies. The intuition for this strategy is that small cliques in the real world are a typical substructure of anomalies, with much more closely linked within the cliques than the mean [109]. Thus for a dataset, we can specify the size of the cliques (i.e., the number of nodes) $p$ and its amount $q$ for anomaly generation. Specifically, randomly sample $p$ nodes from the graph making them fully connected and labeled as anomaly nodes. We iteratively repeat the above process $q$ times to inject a total of $p \times q$ anomalies. Finally, we control the number of injected anomalies according to the size of the dataset. In particular, we fix $p = 15$ and $q = 10, 15, 20, 5, 5, 20$ on BlogCatalog, Flickr, ACM, Cora, Citeseer, and Pubmed, respectively. On the other hand, for attribute perturbations, we base the schema introduced by [110]. Specifically, for each perturbation target node $v_i$, $k$ nodes are randomly sampled in the graph and their distance from the target node is computed. Then, the node $v_j$ with the largest deviation from the target node $v_i$ is selected, and the attribute $\mathbf{X}_i$ of the node $v_i$ to $\mathbf{X}_j$. We set the number of anomalies of the attribute perturbation to $p \times q$ to maintain the balance of different anomalies. In addition, we set $k = 50$ to ensure that the perturbation magnitude is large enough.

### F.2 Description of Baselines

In our evaluation, we provide a comprehensive comparsion of ARC with various supervised and unsupervised GAD methods. On the supervised side, two classic GNNs are included as well as 3 state-of-the-art (SOTA) models specifically tailored for the GAD task. For supervised models, it is assumed that the labels of both normal and abnormal nodes can be used for model training. Therefore, the main binary classification task is used to identify the anomalies:

- **GCN [61]**, as a seminal model in the field of GNN, is known for its ability to process graph-structured data using neighborhood aggregation, facilitating efficient node feature extraction and representation learning.

- **GAT [62]** incorporates the attention mechanism into the GNN framework to achieve dynamic weighting of node contributions. It optimizes its attention according to different downstream tasks to achieve high-quality node representations.

- **BGNN [67]** is a GNN that combines gradient boost decision trees (GBDT) with GNN for graphs with tabular node features. It utilizes the GBDT to handle heterogeneous features

Table 4: Anomaly detection performance in terms of AUPRC (in percent, mean±std). Highlighted are the results ranked first, second, and third. "Rank" indicates the average ranking over 8 datasets.

| Method | Cora | CiteSeer | ACM | BlogCatalog | Facebook | Weibo | Reddit | Amazon | Rank |
|---|---|---|---|---|---|---|---|---|---|
| Supervised - Pre-Train Only | | | | | | | | | |
| GCN | $7.41_{\pm1.55}$ | $6.40_{\pm1.40}$ | $5.27_{\pm1.12}$ | $7.44_{\pm1.07}$ | $1.59_{\pm0.11}$ | $67.21_{\pm15.20}$ | $3.39_{\pm0.39}$ | $6.96_{\pm2.04}$ | 9.6 |
| GAT | $6.49_{\pm0.84}$ | $5.58_{\pm0.62}$ | $4.70_{\pm0.75}$ | $12.81_{\pm2.08}$ | $3.14_{\pm0.37}$ | $33.34_{\pm9.80}$ | $3.73_{\pm0.54}$ | $15.74_{\pm17.85}$ | 7.3 |
| BGNN | $4.90_{\pm1.27}$ | $3.91_{\pm1.01}$ | $3.48_{\pm1.33}$ | $5.73_{\pm1.47}$ | $3.81_{\pm2.12}$ | $30.26_{\pm29.98}$ | $3.52_{\pm0.50}$ | $7.51_{\pm0.58}$ | 10.5 |
| BWGNN | $7.25_{\pm0.80}$ | $6.35_{\pm0.73}$ | $7.14_{\pm0.20}$ | $8.99_{\pm1.12}$ | $2.54_{\pm0.63}$ | $12.13_{\pm0.71}$ | $3.69_{\pm0.81}$ | $13.12_{\pm11.82}$ | 8.6 |
| GHRN | $9.56_{\pm2.40}$ | $7.79_{\pm2.01}$ | $5.61_{\pm0.71}$ | $10.94_{\pm2.56}$ | $2.41_{\pm0.62}$ | $28.53_{\pm7.38}$ | $3.24_{\pm0.33}$ | $7.54_{\pm2.01}$ | 8.4 |
| Unsupervised - Pre-Train Only | | | | | | | | | |
| DOMINANT | $12.75_{\pm0.71}$ | $13.85_{\pm2.34}$ | $15.59_{\pm2.69}$ | $35.22_{\pm0.87}$ | $2.95_{\pm0.06}$ | $81.47_{\pm0.22}$ | $3.49_{\pm0.44}$ | $6.11_{\pm0.29}$ | 6.1 |
| CoLA | $11.41_{\pm3.51}$ | $8.33_{\pm3.73}$ | $7.31_{\pm1.45}$ | $6.04_{\pm0.56}$ | $1.90_{\pm0.68}$ | $7.59_{\pm3.26}$ | $3.71_{\pm0.67}$ | $11.06_{\pm4.45}$ | 9.0 |
| HCM-A | $5.78_{\pm0.76}$ | $4.18_{\pm0.75}$ | $4.01_{\pm0.61}$ | $6.89_{\pm0.34}$ | $2.08_{\pm0.60}$ | $21.91_{\pm11.78}$ | $3.18_{\pm0.23}$ | $5.87_{\pm0.07}$ | 12.1 |
| TAM | $11.18_{\pm0.75}$ | $11.55_{\pm0.44}$ | $23.20_{\pm2.36}$ | $10.57_{\pm1.17}$ | $8.40_{\pm0.97}$ | $16.46_{\pm0.09}$ | $3.94_{\pm0.13}$ | $10.75_{\pm3.10}$ | 5.8 |
| Unsupervised - Pre-Train & Fine-Tune | | | | | | | | | |
| DOMINANT | $21.35_{\pm0.74}$ | $23.02_{\pm1.55}$ | $22.74_{\pm0.95}$ | $35.79_{\pm0.63}$ | $3.56_{\pm0.15}$ | $77.69_{\pm1.43}$ | $3.84_{\pm0.74}$ | $7.48_{\pm0.46}$ | 4.0 |
| CoLA | $13.91_{\pm5.56}$ | $19.51_{\pm3.73}$ | $8.48_{\pm0.51}$ | $10.43_{\pm1.22}$ | $15.19_{\pm11.04}$ | $8.03_{\pm1.19}$ | $4.07_{\pm0.13}$ | $7.27_{\pm1.13}$ | 5.8 |
| HCM-A | $6.41_{\pm1.33}$ | $4.76_{\pm0.51}$ | $4.41_{\pm0.63}$ | $6.62_{\pm0.14}$ | $2.23_{\pm0.76}$ | $27.20_{\pm5.53}$ | $3.10_{\pm0.19}$ | $5.64_{\pm0.09}$ | 11.9 |
| TAM | $13.62_{\pm0.53}$ | $18.66_{\pm1.41}$ | $58.04_{\pm8.17}$ | $13.90_{\pm0.53}$ | $11.11_{\pm3.20}$ | $16.47_{\pm0.08}$ | $3.93_{\pm0.09}$ | $11.56_{\pm1.80}$ | 4.1 |
| Ours | | | | | | | | | |
| ARC | $49.33_{\pm1.64}$ | $45.77_{\pm1.25}$ | $40.62_{\pm0.10}$ | $36.06_{\pm0.18}$ | $8.38_{\pm2.39}$ | $64.18_{\pm0.55}$ | $4.48_{\pm0.28}$ | $44.25_{\pm7.41}$ | 1.9 |

while the GNN considers the graph structure and significantly improves performance on a variety of graphs with tabular features.

- **BWGNN [6]** has spectral and spatial localized band-pass filters to better handle the "right-shift" phenomenon in anomalies, i.e., the distribution of spectral energy is concentrated at high frequencies rather than at low frequencies.

- **GHRN [37]** is a heterophily-aware supervised GAD method based on graph spectra. By emphasizing the high-frequency components of the graph, the method can effectively cut down inter-class edges, thus improving the overall performance of anomaly detection.

For the unsupervised alternative, we consider 4 representative SOTA GAD methods, each of them belonging to a sub-type: data reconstruction, contrastive learning, hop-based auxiliary goal, or affinity-based auxiliary goal:

- **DOMINANT [4]** combines GCN and deep auto-encoder, and its learning objective is to reconstruct the adjacency matrix and node features jointly. It aims to identify structural and attribute anomalies based on reconstruction errors.

- **CoLA [15]** is a contrastive self-supervised learning for anomaly detection on graphs with node attributes. The framework captures the relationship between each node and its neighborhood substructure in an unsupervised manner by sampling novel pairs of contrasting instances and leveraging the local information of the graph.

- **HCM-A [16]** uses hop-count prediction as a self-supervised task to better identify anomalies by modeling both local and global context information. In addition, HCM-A designs two new anomaly scores and introduces Bayesian learning to train the model to capture anomalies.

- **TAM [17]** is designed based on one-class homophily and local affinity. The learning target of TAM is to optimize the proposed anomaly metric (i.e. affinity) end-to-end on the truncated adjacency matrix.

### F.3 Details of Implementation

**Hyper-parameters.** We select some key hyper-parameters of ARC through random search within specified grids. Specifically, the random search was performed within the following search space:

- Hidden layer dimension: {64, 128, 256, 512, 1024}

- Number of MLP layers: {1, 2, 3, 4}

- Propagation iteration: {1, 2, 3, 4, 5}

- Dropout rate: {0, 0.1, 0.2, 0.3, 0.4, 0.5, 0.6, 0.7, 0.8}

- Learning rate: floats between $10^{-5}$ and $10^{-2}$

Table 5: Performance of ARC and its variants in terms of AUROC.

| Variant | Cora | CiteSeer | ACM | BlogCatalog | Facebook | Weibo | Reddit | Amazon |
|---------|------|----------|-----|-------------|----------|-------|--------|--------|
| ARC w/o A | $80.65_{\pm0.71}$ | $83.35_{\pm0.64}$ | $79.29_{\pm0.16}$ | $73.86_{\pm0.18}$ | $62.80_{\pm2.06}$ | $89.69_{\pm0.17}$ | $54.60_{\pm1.92}$ | $64.76_{\pm2.13}$ |
| ARC w/o R | $37.44_{\pm1.40}$ | $31.52_{\pm0.71}$ | $61.83_{\pm1.16}$ | $49.30_{\pm2.06}$ | $20.38_{\pm9.63}$ | $97.72_{\pm0.59}$ | $52.94_{\pm0.96}$ | $50.15_{\pm0.24}$ |
| ARC w/o C | $47.39_{\pm0.42}$ | $53.98_{\pm0.72}$ | $54.24_{\pm1.32}$ | $60.46_{\pm1.23}$ | $48.86_{\pm0.97}$ | $42.84_{\pm3.01}$ | $51.03_{\pm0.86}$ | $69.02_{\pm0.97}$ |
| ARC | $87.45_{\pm0.74}$ | $90.95_{\pm0.59}$ | $79.88_{\pm0.28}$ | $74.76_{\pm0.06}$ | $67.56_{\pm1.60}$ | $88.85_{\pm0.14}$ | $60.04_{\pm0.69}$ | $80.67_{\pm1.81}$ |

- Weight decay: floats between $10^{-6}$ and $10^{-3}$

**Implementation Pipeline.** We employ a fixed set of hyper-parameters to build a generalist GAD model for all datasets. First, we train all the methods (including baselines and ARC) on the training set $\mathcal{T}_{train}$ with full labels. Then, the methods are evaluated on each dataset from $\mathcal{T}_{test}$ respectively. For feature projection, we employ the PCA algorithm to map the raw features into a fixed space with $d_u = 64$. When the original feature dimension is smaller than the predefined projection dimension $d_u$, we use a random projection (e.g., Gaussian random projection) to upscale the feature into a higher dimensionality and then unify the dimensions into $d_u$ with the projection strategy. For the baselines that require fine-tuning, we further conduct dataset-specific tuning at this stage.

**Metrics.** Following [7, 17, 68], we employ two popular and complementary evaluation metrics for evaluation, including area under the receiver operating characteristic Curve (AUROC) and area under the precision-recall curve (AUPRC). A higher AUROC/AUPRC value indicates better performance. We report the average AUROC/AUPRC with standard deviations across 5 trials.

**Computing Infrastructures.** We implemented the proposed ARC using PyTorch 2.1.2, PyTorch Geometric (PyG) 2.3.1, and DGL 0.9.0. All experiments were performed on a Linux server with an Inter Xeon microprocessor E-2288G CPU and a Quadro RTX 6000 GPU.

## G  Supplemental Experiments

### G.1  Performance Comparison in Terms of AUPRC

In terms of AUPRC, Table 4 gives comprehensive comparative results with consistent observations with the AUROC results. Specifically, we have the following observations. ❶ ARC still demonstrates strong anomaly detection in generalist GAD scenarios without any fine-tuning. Specifically, ARC achieves state-of-the-art performance on five of the eight datasets and demonstrates competitive performance on the remaining datasets. On several datasets, ARC showed significant improvement over the best baseline (e.g., $\uparrow$ 131.1% on Cora, $\uparrow$ 98.8% on Citeseer). ❷ GAD methods that only pre-train specific to a dataset usually result in poor generalization to new datasets. Specifically, existing methods perform very erratically on different datasets, which can be attributed to capturing only specific anomaly patterns. ❸ Using dataset-specific fine-tuning, baseline methods can achieve better performance in most case. However, in some cases the improvement can be small or even negative, demonstrating the limitations of fine-tuning.

### G.2  Effectiveness of Context Sample Number

For all test sets, we varied $n_k$ in the range of 2 to 100 and the results are shown in Fig. 5 and Fig. 9. From the figure, we observe that in most cases the performance of ARC increases with the involvement of more context nodes, which indicates its ability to utilize these labeled normal nodes for context learning. Moreover, even if $n_k$ is very small, ARC can still perform well on most datasets.

### G.3  Detailed Results of Ablation Study

To assess the effectiveness of the key design in the ARC, we conducted an ablation study with three variants of the ARC, 1) **w/o A**: using random projection to replace smoothness-based feature alignment; 2) **w/o R**: using GCN to replace ego-neighbor residual graph encoder; and 3) **w/o C**: using binary classification-based predictor and loss to replace cross-attentive in-context anomaly scoring. As can be seen from Table 5 and Table 6, the two metrics AUROC and AUPRC of ARC achieved the best in all datasets except Weibo dataset. A possible explanation is that the Weibo dataset exhibits a

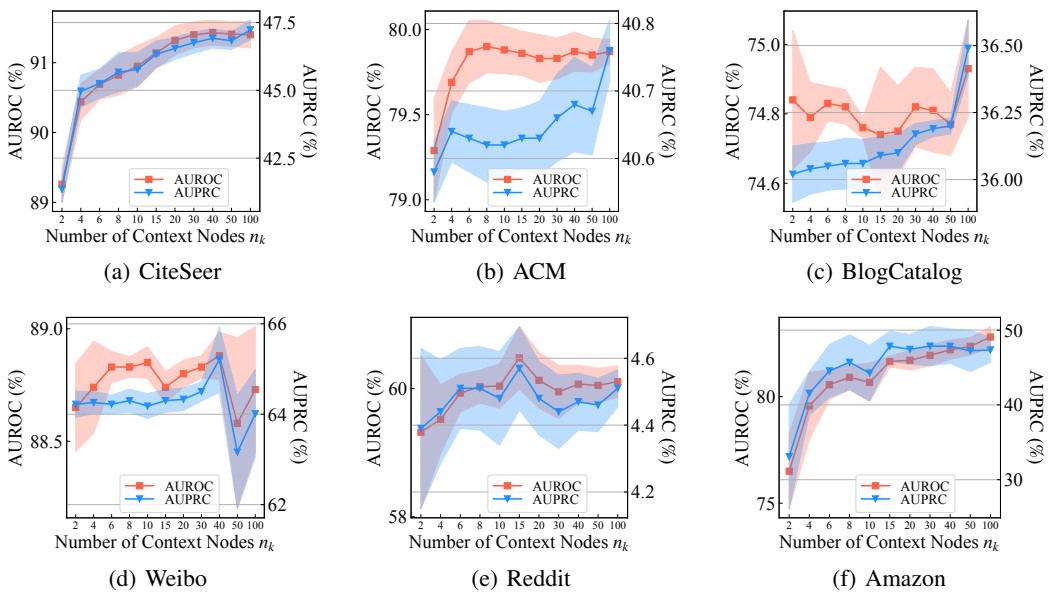

Figure 9: Performance with varying $n_k$ on the rest of six datasets.

Table 6: Performance of ARC and its variants in terms of AUPRC.

| Variant | Cora | CiteSeer | ACM | BlogCatalog | Facebook | Weibo | Reddit | Amazon |
|---|---|---|---|---|---|---|---|---|
| ARC w/o A | $28.51_{\pm1.72}$ | $29.69_{\pm0.68}$ | $29.13_{\pm0.41}$ | $34.23_{\pm0.47}$ | $4.15_{\pm0.33}$ | $67.36_{\pm0.46}$ | $3.62_{\pm0.16}$ | $10.33_{\pm1.61}$ |
| ARC w/o R | $6.98_{\pm0.13}$ | $8.09_{\pm0.30}$ | $4.95_{\pm0.12}$ | $6.32_{\pm0.44}$ | $1.50_{\pm0.21}$ | $92.07_{\pm1.01}$ | $3.59_{\pm0.07}$ | $6.92_{\pm0.19}$ |
| ARC w/o C | $8.21_{\pm0.42}$ | $8.93_{\pm0.67}$ | $16.86_{\pm0.91}$ | $26.87_{\pm0.73}$ | $4.95_{\pm1.50}$ | $12.21_{\pm1.18}$ | $3.74_{\pm0.10}$ | $13.09_{\pm1.31}$ |
| ARC | $49.33_{\pm1.64}$ | $45.77_{\pm1.25}$ | $40.62_{\pm0.10}$ | $36.06_{\pm0.18}$ | $8.38_{\pm2.39}$ | $64.18_{\pm0.55}$ | $4.48_{\pm0.28}$ | $44.25_{\pm7.41}$ |

particular pattern of anomalies. In addition, all three key designs provide significant improvements in performance.

## G.4 Visualization

The attention weights between context nodes and query nodes in other datasets are shown in Fig. 10. As can be seen in Fig. 10, for most of the datasets, the "single-class normal" case in Fig. 4 (a) is met: ARC tends to assign a uniform attention weight to normal nodes. This results in reconstructed embedding that are very similar to the average embedding of context nodes; in contrast, reconstructing anomalies using a combination of a few context nodes results in their embedding being farther from the center. Moreover, corresponding to the "multi-class normal" case in Fig. 4 (b): in Fig. 10 (f), it is observed that each normal query nodes that follow two fixed patterns. In summary, the cross-attention module enables ARC to adapt to various normal/abnormal distribution patterns, conferring it generality.

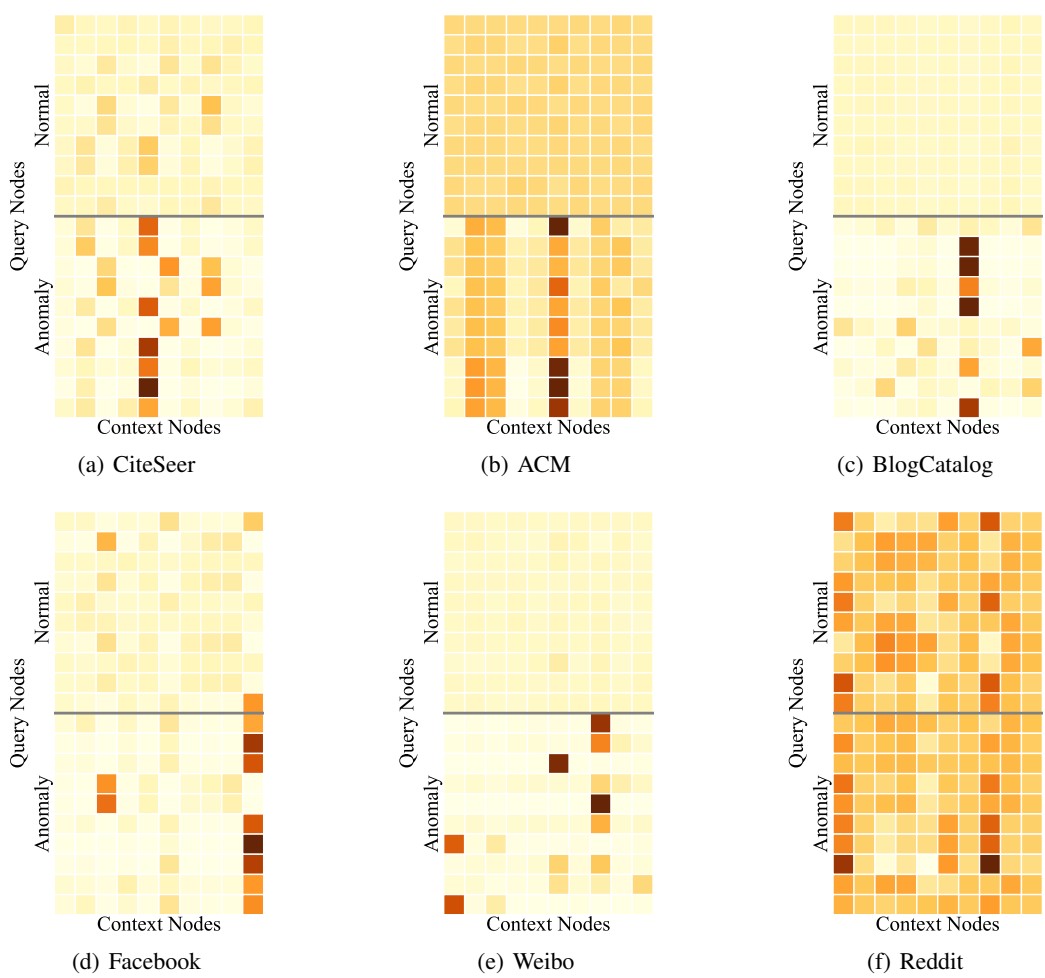

Figure 10: Attention visualization results for more datasets.

