# OpenReview forum: "ARC: A Generalist Graph Anomaly Detector with In-Context Learning"
_NeurIPS.cc/2024/Conference — NeurIPS 2024 poster_

### Official Review · Reviewer_B6wn · 2024-07-03

**Soundness:** 4
**Presentation:** 3
**Contribution:** 4
**Rating:** 7
**Confidence:** 5

**Summary:**

The paper produces a "one-for-all" generalist graph anomaly detection model. The proposed model first use feature projection and smoothness-based ranking to align the features of multiple datasets; then it employs a residual graph neural network to extract representations; finally, a cross-attention module is used to calculate the anomaly scores of query nodes. The paper is well motivated and with sufficient experiments for effectiveness validation.

**Strengths:**

1. The setting of generalist graph anomaly detection is interesting and novel. The research question in this paper has high practice value.
2. The discussion of feature smoothness and abnormality is insightful, which may also inspires future studies.
3. The proposed method is carefully designed with three powerful modules. Each of them plays a critical role in capturing anomaly-aware knowledge and making precise prediction.
4. The experimental results show that the proposed method has significant performance advantage over the baselines.
5. The visualization experiment is interesting, which show the working mechanism of the attention module.

**Weaknesses:**

1. The methodology details of the proposed method are not precise: specifically, which type of feature projection method is used in the experiment should be given.
2. The definitions of the concept of single-class normal and multi-class normal are missing, which hinders the readers to understand the design motivation in Sec. 4.3.
3. The figure is not readable enough. More annotations can be provided in Figure 2, especially for the R and C blocks.

**Questions:**

1. In Eq. 6, the anomaly label is denoted by a bold letter y. Does that mean it's a vector? However, should it be a value actually?
2. What is the reason for the unstable anomaly detection performance under different number of context nodes?

**Limitations:**

I believe the authors have addressed the limitation, and I haven't found any negative societal impact of this work.

---

> ### Author Rebuttal · Authors · 2024-08-06
>
> We appreciate Reviewer B6wn for the insightful feedback and acknowledge our contributions. The responses to the reviewer are as follows.
>
> **Q1: Which type of feature projection method is used in the experiment should be given**
>
> **A1:** Thanks for your valuable suggestion. In our experiments, we used a dimensionality reduction method based on principal component analysis (PCA). We will re-emphasize this in the manuscript.
>
> **Q2: The definitions of the concept of single-class normal and multi-class normal**
>
> **A2:** We appreciate your suggestion. We will add the following definitions in the revised manuscript:
>
> **Dataset with single-class normal:** In this type of dataset, the normal samples share the same pattern or characteristics. For example, in a network traffic monitoring system dataset, normal behavior might be defined by regular patterns of data packets exchanged between a specific set of IP addresses. Any deviation from this single, well-defined pattern, such as an unexpected spike in data volume or communication with unknown IP addresses, can be flagged as anomalous.
>
> **Dataset with multi-class normal:** In this type of dataset, the normal samples are divided into multiple classes, each with distinct patterns or characteristics. For example, in a corporate email communication network dataset, normal data might be defined by regular patterns of email exchanges within specific departments, such as HR, IT, and Finance. Any deviation from these well-defined patterns, such as a sudden spike in emails between normally unconnected departments or an unusual volume of emails from an individual employee to external addresses, can be detected as anomalous.
>
> **Q3: More annotations can be provided in Figure 2**
>
> **A3:** Thanks for the valuable comment! We will add more annotations into Figure 2, including the annotations of $\mathbf{X}$, $\mathbf{R}$, and $\mathbf{H}$ in the "R" block, as well as the annotations of $\mathbf{Q}$, $\mathbf{K}$, $\mathbf{H}_q$, and $\tilde{\mathbf{H}}_q$ in the "C" block.
>
> **Q4: In Eq. 6, the anomaly label is denoted by a bold letter y. Does that mean it's a vector? However, should it be a value actually?**
>
> **A4:** Thanks for your careful review. In fact, $\mathbf{y}_{i}$ is the label of sample (node) i, which is a specific value, and $\mathbf{y}$ is a vector consisting of the labels for all samples.
>
> **Q5: The reason for the unstable performance under different number of context nodes**
>
> **A5:** Thanks for the valuable question! The instability in anomaly detection performance with varying numbers of context nodes can be attributed to several factors:
>
> **Contextual Information Variability:** The number of context nodes directly influences the amount of contextual information available for anomaly detection. With too few context nodes, the model may lack sufficient information to accurately distinguish between normal and anomalous nodes. Conversely, with too many context nodes, the model might include irrelevant or noisy information, which can degrade performance.
>
> **Dataset Characteristics:** Different datasets have varying structures and anomaly distributions, which can affect how context nodes contribute to anomaly detection. The optimal number of context nodes may vary depending on the specific characteristics of each dataset.

---

> > ### Comment · Reviewer_B6wn · 2024-08-14
> >
> > Dear Authors,
> > Thanks for the detailed responses, which fully addresses my previous concerns. I decide to maintain my positive scoring for this paper.

---

### Official Review · Reviewer_VWTk · 2024-07-10

**Soundness:** 4
**Presentation:** 3
**Contribution:** 3
**Rating:** 7
**Confidence:** 4

**Summary:**

This paper proposes a new framework for graph anomaly detection called ARC. ARC uses in-context learning to detect anomalies across various graph datasets without requiring retraining or fine-tuning. It leverages few-shot normal samples during inference to achieve superior performance in anomaly detection tasks.

**Strengths:**

Overall, this paper explores an interesting research topic with a well-designed method. The experimental results are also promising.

1. Methodology design. The design of the proposed method ARC is reasonable, where each component address a challenge in generalized graph anomaly detection.

2. Promising results. The experimental results show that indeed the ARC model can achieve better performance on a series of datasets.

**Weaknesses:**

However, I have several concerns about this paper.

1. In the residual GNN component, a shared MLP is used for feature transformation, which is not vey commonly seen in other GNNs. Could you provide more intuition behind for explanation?

2. In attention mechanism, "value" is an important component in the computational process. However, the in-context learning module doesn't include the computation of "value". Can you give more detailed intuition for this design?

**Questions:**

I still have some minor questions.

1. How to define the projected dimension $d_u$?

2. Considering the cost of fine-tuning (figure 6), will it require a lot of epochs to get the fine-tuned results for the baseline methods?

**Limitations:**

The limitations are upfront in the paper.

---

> ### Author Rebuttal · Authors · 2024-08-06
>
> We appreciate Reviewer VWTk for the valuable feedback and acknowledge our technical contributions and the effectiveness of the proposed method. We address the concerns raised by the reviewer as follows.
>
> **Q1:Why Use a shared MLP in the encoder**
>
> **A1:** Thanks for the insightful comment! According to the empirical analysis in the previous studies [\*3,\*4], GNNs benefit mainly from propagation and require only a small number of MLPs to increase their nonlinear capabilities. Therefore, we increase the nonlinear capability of the propagating features for different neighborhoods by a shared MLP. In addition, the design of the shared MLP allows the model parameters to be drastically reduced, especially when the number of propagations is large.
>
> *[\*3] Wu F, Souza A, Zhang T, et al. Simplifying graph convolutional networks[C]//International conference on machine learning. PMLR, 2019: 6861-6871.*
>
> *[\*4] Nt H, Maehara T. Revisiting graph neural networks: All we have is low-pass filters[J]. arXiv preprint arXiv:1905.09550, 2019.*
>
> **Q2: Reason for value-based attention mechanism**
>
> **A2:** Thanks for the valuable question! As shown in the Discussion in Sec. 4.3 of the paper, in ARC we follow a basic assumption: normal query nodes have similar patterns to several context nodes, and hence their embeddings can be easily represented by the linear combination of context node embeddings. Therefore, the node anomaly score is obtained based on the error before and after the reconstruction. In this case, it is necessary to ensure that the reconstructed features are in the same embedding space as the original features, so we discard the value-based attention. We will add this explanation to the paper.
>
> **Q3: How to define the projected dimension?**
>
> **A3:** Thanks for your insightful comment. The projection size of our ARC model is flexible and can be determined based on a balance between computational efficiency and the ability to capture the underlying anomaly-related features. Specifically, we perform a grid search over a range of dimensions and select the one that yields the best loss on a training set. This approach ensures that the chosen dimension is optimal for capturing the necessary information while maintaining computational feasibility.
>
> **Q4: Will it require a lot of epochs to get the fine-tuned results for the baseline methods?**
>
> **A4:** We appreciate your insightful question. The cost of fine-tuning for baseline methods indeed varies depending on the complexity of the model and the dataset. In our experiments, we observed that baseline methods generally require a significant number of epochs to achieve optimal performance, which contributes to their high training costs. In contrast, our ARC model leverages in-context learning with few-shot normal samples, allowing it to adapt to new datasets without extensive retraining or fine-tuning. This results in a more efficient and cost-effective approach to graph anomaly detection.

---

> > ### Comment · Reviewer_VWTk · 2024-08-14
> >
> > Thanks for the rebuttal. After reading all the comments, I decided to keep my rating of accept.

---

### Official Review · Reviewer_bLtA · 2024-07-10

**Soundness:** 3
**Presentation:** 3
**Contribution:** 3
**Rating:** 8
**Confidence:** 4

**Summary:**

This paper investigates the research problem of "generalist graph anomaly detection (GAD)", targeting to address the cost and generalizability issues of the conventional GAD paradigm. This paper proposes ARC, a "one-for-all" GAD model that is pre-trained on a group of datasets and able to detect anomalies on new datasets on-the-fly. To verify the performance of ARC, experiments on GAD benchmark datasets are executed. The main contributions of this paper are: 1. Addressing a new research problem (generalist GAD); 2. Developing a novel method; 3. Conducting experiments for GAD performance validation.

**Strengths:**

Originality: the proposed method has great originality. The three components in ARC are fresh and novel. The authors further discuss the difference and connection between ARC and other method, which is sufficient to show the originality of this paper.

Quality: This paper is of high quality. The research problem and proposed method are both solid. Comprehensive experiments and discussions are given in the paper (some are in the appendix).

Clarity: The presentation of this paper is good, making it easy to read. Most of the equations, description, figures, and tables are good to understand.

Significance: I'm not an expert of GAD domain, but I think this paper make good significance in the community. Considering the significant difference between the proposed generalist GAD paradigm and the existing one, I think think paper is an important step towards AGI.

**Weaknesses:**

Methodology details: Some specific details of the method design are not clearly given. In specific:

1. For the feature projection, what if the original feature dimension is smaller than the predefined projected dimension?

2. Why doesn't ARC use the value-based cross attention? Is there any disadvantage?

Experiment details: Although the experiments are extensive, I think more discussions should be given for some notable results. In specific:

1. Why unsupervised methods perform better than the supervised methods?

2. Why Figure 5 (b) has a dropping trend at nk=20?

**Questions:**

The questions listed in Weaknesses are expected to be answered. Meanwhile, I have some extra questions:

1. Why "pre-train & fine-tune" settings only include unsupervised methods?

2. Will ARC be time-costly when the number of context nodes become large?

**Limitations:**

From my perspective, the authors have addressed the limitations and the paper seems doesn't have potential negative impact.

---

> ### Author Rebuttal · Authors · 2024-08-06
>
> We appreciate Reviewer bLtA for the positive review and constructive comments. We provide our responses as follows.
>
> **Q1: If the original feature dimension is smaller than the predefined projected dimension**
>
> **A1:** Thanks for the thoughtful comment! When the original feature dimension is smaller than the predefined projection dimension, we use a random projection (e.g., Gaussian random projection) to upscale the feature into a higher dimensionality and then unify the dimensions into $d_u$ with the projection strategy. We will add these implementation details to the manuscript.
>
> **Q2: The reason for using non-value-based cross attention**
>
> **A2:** Thanks for the valuable question. As shown in the Discussion in Sec. 4.3 of the paper, in ARC we follow a basic assumption: normal query nodes have similar patterns to several context nodes, and hence their embeddings can be easily represented by the linear combination of context node embeddings. Therefore, the node anomaly score is obtained based on the error before and after the reconstruction. In this case, it is necessary to ensure that the reconstructed features are in the same embedding space as the original features, so we discard the value-based attention. We will add this explanation to the paper.
>
> **Q3: Why unsupervised methods perform better than the supervised methods**
>
> **A3:** We appreciate your insightful question. The possible reasons are two-fold. Firstly, most supervised GAD methods rely on the binary classification paradigm. This means that supervised GAD is trained based on specific anomalies in the training data, which may result in overfitting the anomaly patterns of the trained dataset. In addition, supervised GAD cannot be fine-tuned on the test set using "normal" samples, which also limits its generalizability to new-coming datasets.
>
> **Q4: Why Figure 5 (b) has a dropping trend at $n_k$=20?**
>
> **A4:** Thanks for your careful review. As in-context node information is variable, the performance of anomaly detection can be affected by the quality of context node information. In our experiments, we sample normal nodes randomly, so there is randomness due to sample-specific sampling bias. This randomness leads to instability in the model's performance at a particular number of context nodes. Despite this, we can still observe that the overall trend of performance is gradually increasing with increasing $n_k$.
>
> **Q5: Why "pre-train & fine-tune" only includes unsupervised methods**
>
> **A5:** Thanks for your insightful comment. To fine-tune the supervised GAD methods on the target dataset, they require both labeled normal samples and anomalies. However, the labeled anomalous samples are not available in our test scenario for fine-tuning by supervised methods. On the other hand, unsupervised methods are label-free and thus can be fine-tuned on the testing dataset.
>
> **Q6: Does large number of context nodes cause ARC to become time-costly?**
>
> **A6:** Thanks for the valuable question. As can be seen from the section Complexity Analysis in Appendix E.2 of the paper, the model inference mainly consists of two parts, node embedding generation and anomaly scoring, $\mathcal{O}(L(md_u + nd_uh + nh^2))$ and $\mathcal{O}(n_q(n_k+1)h)$, respectively. It is worth noting that $n_k \ll n$, thus complexity mainly depends on the node embedding generation. Therefore, a larger number of context nodes does not make ARC time-costly.
>
> In addition, we compare the test times (in second) of ARC for datasets with different numbers of context nodes, and the results are shown in the table below. We can see that the effect of different numbers of context nodes on ARC is almost negligible, indicating the high running efficinecy of ARC when $n_k$ is large.
>
> |$n_k$|**200**|**400**|**1000**|**2000**|
> |:-|:-|:-|:-|:-|
> |ACM|0\.0301|0\.031|0\.0319|0\.0344|
> |Facebook|0\.0071|0\.0075|0\.0072|0\.0072|
> |Reddit|0\.0224|0\.0224|0\.0236|0\.025|
> |citeseer|0\.011|0\.0101|0\.0102|0\.0102|

---

> > ### Comment · Reviewer_bLtA · 2024-08-13
> >
> > Thanks for the authors' rebuttal, which addresses all my concerns. I would like to raise my rating in response to the authors' efforts during the rebuttal.

---

### Official Review · Reviewer_HCAf · 2024-07-15

**Soundness:** 2
**Presentation:** 3
**Contribution:** 2
**Rating:** 4
**Confidence:** 4

**Summary:**

This paper proposed AARC, a generalist GAD approach to detect anomalies across various graph datasets on the fly. It consists of feature alignment module, residual graph encoder and in-context anomaly scoring module.

**Strengths:**

1. The experimental results show that the proposed method outperform most of the baseline methods across various datasets.
2. The presentation of this paper is good and the paper is easy to follow.

**Weaknesses:**

1. The smoothness-based feature sorting is done in the raw input feature. In this case, after unify the feature dimensionality with the feature projection, how do you smooth the new feature $\hat{X}^{(i)}$ with smoothness-based feature sorting? In another word, how do you aligns dimensionality and smooth feature at the same time?
2. In line 232 and 233, the authors mention that "In the first two steps, we perform propagation on the aligned feature matrix $X'=X^{[0]}$ for $L$ iterations, and then conduct transformation on the raw and propagated features with a shared MLP network". This is confusing. It's unclear weather the input feature is aligned feature matrix or the raw feature. Besides, the aligned feature matrix should be $\tilde{X}$ rather than $X$.
3. Equation 3 can be written as $Z^{[l]}=MLP(\tilde{A}X^{[l-1]})=\sigma(\tilde{A}X^{[l-1]}W)$, where $W$ is the weight matrix of MLP and $\sigma$ is the activation function. In this case, what's the difference between equation 3 and traditional GNN. The claim of equation 3 to capture the high-frequency signals and local heterophily is not convincing.
4. The design of cross-attentive in-context anomaly scoring is based on the assumption that the normal query have similar patterns to several context nodes and their embeddings can be easily represented by linear combination of context node embeddings. However, in many cases, given the limited number of  normal nodes, especially for the test graph, it's highly likely that some unsampled normal nodes cannot be represented by linear combination of other normal nodes. For instance, the sampled normal nodes from class 1 to class 5, while there exist some unsampled normal nodes from class 6. In this case, these unsampled normal nodes may not be represented by the linear combination of other normal nodes and these nodes tend to have large reconstruction error, thus being labeled as the abnormal nodes. How do you mitigate this issue?

**Questions:**

1. The smoothness-based feature sorting is done in the raw input feature. In this case, after unifying the feature dimensionality with the feature projection, how do you smooth the new feature $\tilde{X}^{(i)}$ with smoothness-based feature sorting? In another word, how do you aligns feature dimensionality and smooth feature at the same time?
2. The design of cross-attentive in-context anomaly scoring is based on the assumption that the normal query have similar patterns to several context nodes and their embeddings can be easily represented by linear combination of context node embeddings. However, in many cases, given the limited number of  normal nodes, especially for the test graph, it's highly likely that some unsampled normal nodes cannot be represented by linear combination of other normal nodes. For instance, the sampled normal nodes from class 1 to class 5, while there exist some unsampled normal nodes from class 6. In this case, these unsampled normal nodes may not be represented by the linear combination of other normal nodes and these nodes tend to have large reconstruction error, thus being labeled as the abnormal nodes. How do you mitigate this issue?
3. For the supervised baseline methods, what is the available label information for the test graphs. Based on the setting in the paper, it seems that only a small percentage of normal nodes are available for training for test graphs. In this case, how do you train these supervised baseline methods? What is the training procedure for these baseline methods?

**Limitations:**

The authors list limitations in conclusion.

---

> ### Author Rebuttal · Authors · 2024-08-06
>
> We are grateful to Reviewer HCAf for providing insightful feedback. The detailed responses are provided below.
>
> **Q1: Details of feature alignment with sorting**
>
> **A1:** Thank you for the thoughtful comment. In ARC, the smoothness-based feature sorting is performed on the projected features rather than the raw features (please see Line 209 and Algorithm 1 for details). In this context, the projected features of different datasets can be aligned in the same order. We understand that Equation (2) may mislead readers into thinking that $s_k$ is calculated from raw features; but actually, it can be calculated from projected features as well. We will further emphasize this to prevent such misunderstandings.
>
> **Q2: Confusing statement of "raw features"**
>
> **A2:** We appreciate the reviewer for pointing out the confusing sentence! This is a wrong statement. The right process is that "the MLP transformation is conducted on the **aligned** and propagated features", rather than the **raw** features. Here we denote the raw, projected, and aligned features as $\mathbf{X}$, $\tilde{\mathbf{X}}$, and $\mathbf{X}'$, respectively, so $\mathbf{X}^{[0]}=\mathbf{X}'$. Algo. 1 and Algo. 2 can better demonstrate the operation of this part. We thank the reviewer again for carefully finding this issue and will definitely fix it in the later version.
>
> **Q3: How the encoder capture high-frequency signals**
>
> **A3:** Thanks for the insightful comment. In ARC, Eq.(3) is similar to a traditional GNN and can be viewed as low-pass filters. However, with the first sub-formula in Eq.(4), i.e., $\mathbf{R}^{[l]} = \mathbf{Z}^{[l]} - \mathbf{Z}^{[0]}$, the final residual representation $\mathbf{R}^{[l]}$ can capture high-frequency signals. Specifically, if we omit the non-linear transformation in MLP, then $\mathbf{R}^{[l]}=\tilde{\mathbf{A}}^l\mathbf{X}\mathbf{W} -  \mathbf{X}\mathbf{W}=(\tilde{\mathbf{A}}^l-\mathbf{I})\mathbf{X}\mathbf{W}$. Here, $(\tilde{\mathbf{A}}^l-\mathbf{I})$ can typically serve as a high-pass filter according to [\*1, \*2]. Especially when $l=1$, $(\tilde{\mathbf{A}}-\mathbf{I})$ can serve as Laplacian graph filter, a classic high-pass filter. A detailed discussion is also given in the final paragraph of Appendix C. We hope this answer can address the reviewer's concern.
>
> *[\*1] Zhu, Meiqi, et al. "Interpreting and unifying graph neural networks with an optimization framework." WebConf'21.*
>
> *[\*2] Bo, Deyu, et al. "Beyond low-frequency information in graph convolutional networks." AAAI'21.*
>
> **Q4: Few-shot setting meets unseen normal class**
>
> **A4:** Thanks for raising this valuable question! Theoretically, the unseen classes may lead to confusing predictions; however, in practice, we find that this issue does not affect the performance a lot. Taking the performance on Cora dataset (which has 7 normal classes) as an example (see Fig. 5 (a)), when shot number $n_k$ is smaller than 7, which means the labeled samples must not cover all classes, the performance is still acceptable and significantly better than the baselines in Table 1. Potential reasons are: 1) the difference between anomalies and normal samples is significantly larger than the intra-class difference of normal samples; 2) the normal samples belonging to unseen classes can be easily represented by the few-shot normal samples.
>
> However, to mitigate the potential negative impact of this issue, we can also employ the following extra strategies: 1) Increase the number of few-shot normal samples as much as possible to cover all normal classes; 2) Enhance the diversity of the few-shot normal samples; 3) Integrate active learning techniques into ARC, which allows the model to select the most confusing sample to be annotated.
>
>
> **Q5: Implementation details of supervised baselines**
>
> **A5:** Thanks for the valuable question. As the reviewer mentioned, the supervised baselines require both labeled normal samples and anomalies for training. However, in the setting of generalist anomaly detection, only few-shot normal samples are available, making it difficult to fine-tune the supervised baselines on the testing datasets. Hence, we only pre-train them on the training datasets and directly apply the pre-trained models to the testing datasets. Note that, in the datasets for pre-training, both labeled anomalies and normal samples are available, ensuring effective pre-training for supervised baselines and ARC. We will add the implementation details to the revised manuscript.

---

> > ### Comment · Reviewer_HCAf · 2024-08-11
> > **Reply to authors' rebuttal**
> >
> > I thank the authors for their thoughtful rebuttal. Most of my concerns have been properly addressed, but some remain:
> > - For the answer to question 4, though the experimental results in Table 1 on the Cora dataset show that the proposed methods significantly outperform the other baseline methods, it can be also observed that some baseline methods can outperform ARC on other datasets, such as Facebook, Weibo. Specifically, TAM (Unsupervised - Pre-Train & Fine-Tune) outperforms ARC by around 7% on Facebook dataset. It's not a consistent observation throughout all evaluated datasets. Thus, I am not fully convinced by the statement that this issue does not affect the performance a lot.
> > - I look at the code of ARC. The forward function in ARC class (Line 21 -Line 39 in model.py file) corresponds to the implementation of Ego-Neighbor Residual Graph Encoder, I find that **the adjacency matrix is not used for representation learning, which is not consistent with Equation 3**.

---

> ### Author Response · Authors · 2024-08-11
> **Reply to the further questions raised by Reviewer HCAf**
>
> We are grateful to Reviewer HCAf for the valuable feedback.
> Please find our responses to your new questions below. We hope that our response addresses your concerns.
>
> **Q6: Impact of unseen normal class problem**
>
> **A6:** We appreciate you once again for highlighting the unseen normal class problem. We agree that a deeper discussion on the impact of this issue is necessary. As shown in Fig. 5 and Fig. 9, the performance gaps between $n_k=2$ (where normal classes should not be fully covered) and $n_k=100$ (where normal classes should be covered) are generally within the range of $0.5$% - $5$% in terms of AUROC. This indicates that the unseen normal class problem may be a factor affecting the performance of ARC. However, we would like to note that the unseen normal class problem is not a fatal issue of ARC, since the performance when $n_k=2$ remains generally acceptable and won't collapse to indistinguishable prediction (e.g., AUROC=50% and AUPRC=0%).
>
> We want to express our appreciation to the reviewer once again for identifying the potential unseen normal class problem in ARC. We will definitely discuss the potential limitations caused by this issue in the revised manuscript and provide possible strategies (in A4) for users/readers to mitigate this problem.
>
> **Q7: Codes of propagation in ARC**
>
> **A7:** Thank you for your detailed observation regarding the implementation of the Ego-Neighbor Residual Graph Encoder in the ARC model.
>
> The reason the adjacency matrix is not directly used in the forward function (Lines 21-39 in model.py) is due to our decoupled architecture in Eq.(3), i.e., the propagation is directly conducted on the features without learnable parameters. In this case, we can move the propagation step into the codes of data preprocessing to enhance the running efficiency (since we can only run it once). Specifically, the codes of propagation are in the "propagated" function (Lines 143-148 in utils.py), which is called in Lines 53-57 in main.py before the training of the encoder. Then, the propagated features (saved in Dataset.graph.x_list) are subsequently used in the forward function (Lines 21-39 in model.py) for downstream residual encoding.
>
> We appreciate your suggestion, and to prevent potential misunderstandings, we will incorporate more comments in the publicly released code for interpretation.

---

> ### Comment · Reviewer_HCAf · 2024-08-13
> **Reply to Authors' response**
>
> Thanks for the detailed reply. I am still concerned about the generalization of the proposed method as the generalization capability is the major contribution in this paper. I agree with the comment by reviewer **RwhU** that "The method section did not provide a solid theoretical analysis to support this claim." when dealing with the new dataset. In addition, the experimental results are not sufficient to demonstrate the capability of the proposed method on the unseen tasks.
>
> In addition, I have one more question regarding your response to Reviewer **RwhU** about the results of few-shot samples. Why does the performance of ARC achieve the best performance with limited number of pseudo samples (e.g.,
> #Pseudo Normal = 2) on some datasets, such as Amazon, Facebook and Weibo? As you mention during the rebuttal that "the advantage of having a larger number of normal samples reduces the impact of anomalies within the pseudo normal samples.", while the performance of ARC tends to decrease on these datasets when the number of pseudo normal samples increases. Do you have any explanation or thoughts to this observation?

---

> > ### Author Response · Authors · 2024-08-14
> >
> > We genuinely appreciate your thoughtful questions. Before we address your subsequent questions, could you confirm if our response has addressed your concerns regarding **Q1-Q7**? We want to ensure that all your concerns have been adequately addressed before we discuss any new questions. If there are any aspects you feel need further clarification or improvement, please do not hesitate to let us know.
> >
> > The responses to your new questions are given as follows.
> >
> > **Q8: Generalization capability of ARC**
> >
> > **A8:** From a theoretical perspective, ARC is equipped with an in-context learning (ICL) technique, which enhances its generalization capability on new datasets. Specifically, the ICL mechanism enables ARC to distinguish between normal and abnormal samples by using a set of normal samples from each given testing dataset, rather than fitting to the anomaly patterns of a specific training dataset. Following the assumption that normal samples are more similar to each other than anomalies, the cross-attention-based ICL module can learn to make accurate predictions by leveraging inter-sample similarity on new datasets.
> >
> > From an experimental perspective, we have made every effort to demonstrate the generalizability of ARC through experiments on datasets from a wide range of domains. Specifically, in the original manuscript, we included 8 datasets from 3 domains: **citation network, social network, and co-review network**. During the review process, we further expanded our experiments to cover 4 additional domains: **co-author network, co-purchase network, transaction record network, and work collaboration network**. It is important to note that our model had **never seen** data from these 4 new domains during training, providing strong empirical evidence of ARC’s generalization capabilities on diverse and unseen tasks. To the best of our knowledge, we have covered datasets from most of the mainstream domains in graph anomaly detection (GAD), including **4 datasets for training and 12 datasets for testing**, which exceeds the evaluation scope of the vast majority of GAD studies.
> >
> > **Q9: Results discussion of experiments of pseudo normal trick**
> >
> > **A9:** Regarding your question about the performance of ARC with a limited number of pseudo samples, we would like to clarify the following points:
> > **Optimal number of pseudo samples.** The statement “Why does the performance of ARC achieve the best performance with limited number of pseudo samples (e.g., #Pseudo Normal = 2) on some datasets, such as Amazon, Facebook and Weibo?” is not accurate. As mentioned in our response to Reviewer RwhU, ARC achieves its best performance on these datasets with #Pseudo Normal = 10. This is because, in real-world scenarios, the number of normal samples typically far exceeds the number of anomalies. Therefore, with a sample size of 10, the probability of selecting true normal samples is higher, which is why we set the default sample size to 10 in all our experiments. This demonstrates that ARC can achieve competitive performance with a small number of shot samples.
> >
> > **Impact of sample size on performance.** The statement "the advantage of having a larger number of normal samples reduces the impact of anomalies within the pseudo normal samples" was intended to illustrate the intuitive feasibility of the "pseudo normal trick." However, this does not imply that increasing the number of pseudo normal samples will always enhance ARC’s performance. On the contrary, when the sample size is very small (e.g., $n_k$=2 or 10), the probability of selecting true normal samples is higher. As the sample size increases, the likelihood of including false normal samples also increases (especially when the anomaly ratio of the dataset is relevantly large), introducing more noisy psuedo samples and potentially degrading performance. To sum up, the performance of ARC with the "pseudo normal trick" is influenced by the "purity" of the sampled pseudo examples, which in turn depends heavily on the anomaly ratio of the dataset and experimental randomness. Therefore, it is reasonable that different datasets exhibit varying preferences for the pseudo sample size.
> >
> > Last but not least, we'd like to clarify that **the exploration of the pseudo-label trick is merely an additional exploratory experiment**. We aimed to investigate the possibility of applying ARC to a purely unsupervised (zero-shot) scenario. Surprisingly, the results show that the performance is still acceptable, indicating ARC's potential. However, our main focus remains on the few-shot setting, where reliable content samples are available for ICL. We hope this experiment demonstrates the potential of ARC and inspires future work in this direction, but it is not the main contribution of this paper.

---

> > > ### Author Response · Authors · 2024-08-14
> > >
> > > Finally, we express our sincere appreciation for your thorough review and insightful questions. We want to emphasize that this paper introduces a new paradigm in graph anomaly detection (GAD), known as generalist GAD, which presents a greater challenge than conventional GAD problems. Additionally, the proposed method is fundamentally different from traditional GAD models, incorporating well-crafted techniques for data alignment and in-context learning. We hope that this frontier discovery of “one-for-all” GAD models will inspire the development of innovative and more generalized solutions within the community.
> > >
> > > We kindly ask you to review our explanations with an open mind, considering how our method diverges from conventional GAD approaches. Your expertise is invaluable, and we would be grateful if you could reassess our work in light of these clarifications.
> > >
> > > Given this new perspective, we respectfully request that you consider providing a fair evaluation that reflects the true nature and potential impact of our research. Your impartial assessment would be greatly appreciated.
> > >
> > > Thank you for your time and consideration.

---

### Official Review · Reviewer_RwhU · 2024-07-24

**Soundness:** 3
**Presentation:** 3
**Contribution:** 3
**Rating:** 4
**Confidence:** 4

**Summary:**

The paper introduces an ARC method designed to detect anomalies across diverse graph datasets without the need for dataset-specific training. ARC leverages in-context learning with few-shot normal samples during inference, comprising three main components: a smoothness-based feature alignment module, an ego-neighbor residual graph encoder, and a cross-attentive in-context anomaly scoring module. The model demonstrates superior performance, efficiency, and generalizability compared to existing GAD methods, as validated by extensive experiments on multiple benchmark datasets.

**Strengths:**

1. Graph anomaly detection is very challenging. This paper untangles this problem into three folds and solves them one by one, making the paper technically sound.

2. The proposed ARC model is a “one-for-all” GAD model that is capable of identifying anomalies across target datasets from diverse domains, without the need for re-training or fine-tuning.

**Weaknesses:**

1. The paper's claim that it is "capable of detecting abnormal nodes across diverse graph datasets from various application domains without any training on the specific target data" is a bit ambitious. The method section did not provide a solid theoretical analysis to support this claim.

2.  The statement in Line 102 should be moved to the introduction section to declare the scope of this work.

3. The datasets used in the experiments are somewhat similar. It is recommended to include more diverse datasets from different backgrounds.

4. The model relies on the availability of a few-shot normal samples for in-context learning. In some practical situations, obtaining these samples may be challenging, particularly in highly anomalous or unknown environments.

**Questions:**

see the weaknesses above

**Limitations:**

yes

---

> ### Author Rebuttal · Authors · 2024-08-06
>
> We appreciate Reviewer RwhU for the perception of our contributions and thank the reviewer for the insightful feedback. The detailed responses are provided below.
>
> **Q1: Ambitious statement of the learning target**
>
> **A1:** Thank you for raising this valuable comment. While we cannot guarantee that the proposed method will perform well on every new dataset, it is capable of generating reliable predictions through in-context learning on unseen datasets. Our experiments also demonstrate the generalizability of the proposed method. To clarify the capabilities of ARC, we will carefully revise the statement to ensure it is more precise.
>
> **Q2: Declare the scope of this work**
>
> **A2:** Thank you for the insightful suggestion! In the revised manuscript, we will further emphasis the scope of this work (node-level generalist GAD) in the introduction section.
>
> **Q3: Diversity of datasets**
>
> **A3:** We appreciate the reviewer’s valuable suggestion. We recognize the importance of dataset diversity in the generalist GAD setting, and we have included datasets from three different domains (Citation network, Social network, and Co-review network) with both real and injected anomalies. To further evaluate the generalizability of ARC, we have added four additional datasets from four new domains as test datasets for ARC. These new datasets exhibit significant diversity compared to the training datasets, as they come from diverse, new, and unseen domains. The statistics are as follows.
>
>
> |**Name**|**\#Nodes**|**\#Edges**|**\#Dim.**|**Anomaly**|**Anomaly\_Type**|**Domain**|
> |:-|-:|-:|-:|-:|:-|:-|
> |Tolokers|11,758|519,000|10|21\.80%|Real|Crowd-sourcing Service network|
> |T-Finance|39,357|21,222,543|10|4\.60%|Real|Finance network|
> |CoAuthor CS|18,333|163,788|6,805|3\.27%|Injected|Co-author network|
> |Amazon Photo|7,650|238,162|745|5\.88%|Injected|Co-purchase network|
>
>
> The AUROC comparison on the four new datasets is shown in the following table. As seen in the table, the proposed method ARC demonstrates competitive performance on all new datasets, showcasing its strong generalizability to unseen domains. Additionally, ARC exhibits excellent scalability on large datasets (e.g., T-Finance). We hope that these additional experiments on unseen domains and datasets address your concerns about dataset diversity.
>
> |method|CoAuthor CS|Amazon Photo|Tolokers|T-Finance|
> |:-|:-|:-|:-|:-|
> |DOMINANT(PT)|0.6061|0.4739|0.4812|OOM|
> |CoLA(PT)|0.7026|0.5608|0.5236|0.5180|
> |TAM(PT)|0.6995|0.5835|0.5051|OOM|
> |DOMINANT(PTFT)|0.7363|0.6060|0.4744|OOM|
> |CoLA(PTFT)|0.7927|0.6421|**0.5838**|0.2381|
> |TAM(PTFT)|0.6996|0.5728|0.5081|OOM|
> |**ARC**|**0.8273**|**0.7555**|0.5712|**0.6410**|
>
> **Q4: Requirement on few-shot samples**
>
> **A4:** Thank you for the thoughtful comment. In-context learning is a critical mechanism in ARC to ensure the model can receive basic patterns of test datasets. Since we only require a few normal samples, the annotation cost of ARC can be low. In parctice, it is easier to find some normal data in a real-world graph because they are more numerous and their patterns are more representative. Also, according the results in Fig.5, ARC can also work well when the shot number is quite small, which further shows its label efficiency.
>
> In cases where labeled samples are not available, we can introduce a "pseudo normal trick". This involves randomly sampling nodes from the test graph to serve as "pseudo normal samples". Given the assumption that the number of normal samples is significantly larger than that of anomalies, most of the pseudo normal samples will actually be true normal samples. The advantage of having a larger number of normal samples reduces the impact of anomalies within the pseudo normal samples. We can then use these pseudo normal samples as context nodes in ARC.
>
> |*#Pseudo Normal*|**Cora**|**CiteSeer**|**ACM**|**BlogCatalog**|**Facebook**|**Weibo**|**Reddit**|**Amazon**|
> |:-|:-|:-|:-|:-|:-|:-|:-|:-|
> |2|0.8434|0.8991|0.7947|0.7422|0.6842|0.8775|0.5793|0.6321|
> |10|0.8665|0.9195|0.7962|0.7357|0.7049|0.8817|0.5843|0.6459|
> |30|0.8665|0.9206|0.7985|0.7425|0.6938|0.8376|0.5927|0.6412|
> |50|0.8673|0.9208|0.7980|0.7417|0.6949|0.8352|0.5923|0.6380|
> |100|0.8691|0.9213|0.7992|0.7409|0.6877|0.8264|0.5926|0.6420|
> |200|0.8690|0.9212|0.7989|0.7422|0.6843|0.7969|0.5938|0.6439|
>
> The experimental results of using different numbers of pseudo normal samples are shown in the table above. Surprisingly, ARC demonstrates competitive performance with the pseudo normal trick. It is important to note that no labeled data is used at all in this case. However, for several datasets (e.g., Weibo and Amazon), the performance is still inferior to that of the few-shot counterpart, highlighting the importance of in-context normal samples in generalist GAD.

---

> ### Author Response · Authors · 2024-08-13
>
> Dear Reviewer RwhU,
>
> We sincerely appreciate your time and expertise in reviewing our submission. Acknowledging the demands on your schedule, we are mindful not to intrude on your time. However, we would be grateful if you confirm that our rebuttal adequately addresses your concerns.
>
> Thank you in advance for your consideration.
>
> Authors

---

### Author Rebuttal · Authors · 2024-08-06

We sincerely thank all the reviewers for their valuable and insightful comments. We are glad that the reviewers find that the studied problem is novel and significant (Reviewer RwhU, bLtA, VWTk, and B6wn), the proposed method is novel and well-motivated (Reviewer bLtA, VWTk, and B6wn), the empirical studies are adequate and reasonable (Reviewer HCAf, bLtA, VWTk, and B6wn), and the writing is smooth and has a good storyline (Reviewer HCAf, and bLtA).

To the best of our efforts, we provided detailed responses to address the concerns raised by each reviewer in the following. Meanwhile, we carefully revised the paper according to the reviewers’ comments. We will incorporate all the feedback in the final version.

Specifically, the main modifications we made are as follows.
﻿
* We added additional experiments to validate the effectiveness of the proposed method on a wider range of datasets from different domains (see the reply A3 to Reviewer RwhU for details).
* We added additional experiments to discuss the correlation between the time complexity of the proposed method and the number of context samples (see the reply A6 to Reviewer bLtA for details). The discussion shows that the complexity of the method is almost independent of the number of context samples.
* We added additional experiments discussing the robustness of the model in the purely unsupervised scenario, i.e., zero-shot case (see the reply A4 to Reviewer RwhU for details).
* We illustrated the implementation details of the proposed methodology as well as the design intuition of the model (see the replies to Reviewer HCAf, bLtA, VWTk, and B6wn for details).
* We illustrated the implementation details of the baselines (see the replies to Reviewer HCAf and bLtA for details).

---

### Decision · Program_Chairs · 2024-09-25

**Decision:**

Accept (poster)

**Comment:**

This paper proposes a graph anomaly detection approach that identifies anomalies on-the-fly using only few-shot normal nodes. The reviewers appreciated the clear presentation, noting the high originality and generality of the proposed approach, as well as the extensive experiments conducted. Given that several reviewers championed the paper, I recommend acceptance.